# MoCha: Towards Movie-Grade Talking Character Generation

Cong Wei[1,2,*] Bo Sun[2,†] Haoyu Ma[2], Ji Hou[2], Felix Juefei-Xu[2], Zecheng He[2], Xiaoliang Dai[2], Luxin Zhang[2], Kunpeng Li[2], Tingbo Hou[2], Animesh Sinha[2], Peter Vajda[2], Wenhu Chen[1,‡]

[1]University of Waterloo      [2]GenAI, Meta

{cong.wei,wenhu.chen}@uwaterloo.ca

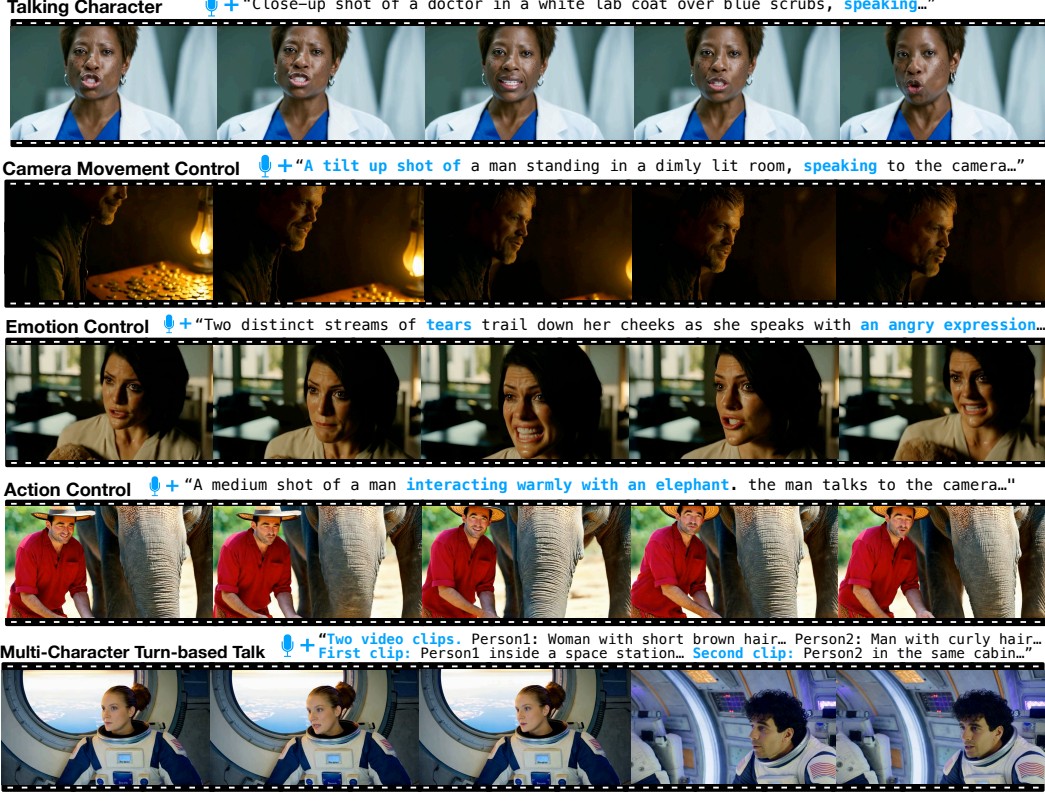

Figure 1: MoCha is an end-to-end dialogue-centric video generation model that takes only **speech** and **text** as input, without requiring any auxiliary conditions. More videos are available at https://congwei1230.github.io/MoCha/

## Abstract

Recent advancements in video generation have achieved impressive motion realism, yet they often overlook character-driven storytelling, a crucial task for automated film, animation generation. We introduce **Talking Characters**, a more realistic task to generate talking character animations directly from speech and text. Unlike talking head, Talking Characters aims at generating the full portrait of one or

---

[*]Work done during the first author's internship at GenAI, Meta

[†]Project Lead

[‡]Corresponding Author

39th Conference on Neural Information Processing Systems (NeurIPS 2025).

more characters beyond the facial region. In this paper, we propose MoCha, the first of its kind to generate talking characters. To ensure precise synchronization between video and speech, we propose a localized audio attention mechanism that effectively aligns speech and video tokens. To address the scarcity of large-scale speech-labelled video datasets, we introduce a joint training strategy that leverages both speech-labeled and text-labeled video data, significantly improving generalization across diverse character actions. We also design structured prompt templates with character tags, enabling, for the first time, **multi-character conversation with turn-based dialogue**—allowing AI-generated characters to engage in context-aware conversations with cinematic coherence. Extensive qualitative and quantitative evaluations, including human evaluation studies and benchmark comparisons, demonstrate that MoCha sets a new standard for AI-generated cinematic storytelling, achieving superior realism, controllability and generalization.

# 1   Introduction

Automating film production holds immense commercial potential, promising to democratize cinematic-level storytelling by enabling content creators to effortlessly generate films through natural language [1–4]. In film, dialogue plays a central role in conveying narratives. Ideally, creators should be able to specify rich storylines involving multiple characters—whether realistic humans or stylized cartoons—that engage in meaningful dialogue, express emotions, and perform full-body actions. Such talking characters serve as powerful mediums for delivering impactful messages, communicating ideas, and engaging audiences. Beyond film, they also enable a wide range of downstream applications, including digital assistants, virtual avatars, advertising, and educational content.

Despite impressive progress in video generation, current video foundation models are primarily designed for narration-style, non-dialogue scenes. Models such as SoRA, Pika, Luma, Hailuo, and Kling [5–12] produce characters with arbitrary lip movements and disconnected facial expressions, lacking control over actual speech content. As a result, these characters appear lifeless and fail to deliver messeage to the audience, limiting the models' applicability in real-life cinematic production.

Meanwhile, speech-conditioned video generation is still in its infancy, primarily focused on simplified talking-head scenarios. Models such as Loopy, Hallo3, and EMO [13–19] are limited to cropped face regions with static cameras, ignoring essential elements such as full-body actions, camera motion, and multi-character interactions. These limitations hinder their characters' expressiveness and make them unsuitable for realistic, engaging storytelling.

To bridge the gap between non-dialogue video generation and constrained talking-head synthesis, we introduce the novel task of **Talking Characters**, which directly targets the goal of automating dialogue-centric film production. The task is defined as generating lifelike digital characters from natural language and speech inputs that express synchronized speech, realistic emotions, and full-body actions under dynamic camera movements (see section 2). To tackle this task, we propose **MoCha**, the first end-to-end Diffusion Transformer (DiT) model designed to produce high-quality, movie-grade talking character videos. MoCha demonstrates compelling storytelling capabilities. As shown on the project website [ahttps://congwei1230.github.io/MoCha/](ahttps://congwei1230.github.io/MoCha/), we lightly edited MoCha-generated clips into a 1-minute, emotionally engaging narrative, illustrating its potential for real-world filmmaking.

MoCha introduces several key technical innovations tailored for this task:

- **First End-to-End DiT Without Auxiliary Control Signals:** MoCha is the first DiT-based model to demonstrate that high-quality lip synchronization and natural character motion can be achieved using only text and speech—without relying on external control signals such as reference images, pose skeletons, or facial keypoints [14, 15, 20, 13, 16, 21, 17]. Contrary to the dominant belief that audio alone is insufficient, we show that strong audio-visual alignment can emerge purely from end-to-end training.

- **Localized Audio Attention**: We propose a attention mechanism tailored for DiT-based dialogue-diven video generation, which addresses the temporal mismatch between compressed video tokens and high-resolution audio inputs (see Sec. 3.2). This design significantly improves lip-sync accuracy and speech-video alignment.

- **Curriculum-Based Multimodal Training:** We introduce a training strategy that combines limited speech-labeled and large-scale text-only video data through a modality-aware curriculum. By mixing multimodal and unimodal supervision and progressively increasing visual complexity, our approach improves convergence, enhances generalization, and enables MoCha's *universal controllability* via natural language prompts—supporting fine-grained control over character expressions, actions, interactions, camera movement, and scene composition without relying on auxiliary signals.

- **Multi-Character Conversational Video Generation:** MoCha is the first model to support coherent, turn-based dialogue among multiple characters—overcoming the single-speaker limitation of prior methods. This capability enables cinematic, story-driven video synthesis with dynamic character interactions.

To evaluate MoCha's performance, we further curated **MoCha-Bench**, a benchmark tailored for Talking Characters generation tasks. Both human evaluations and automatic metrics demonstrate that MoCha set a new standard for talking character video generation and represents a significant step toward achieving controllable, narrative-driven video synthesis, with broad applications in film production, animation, virtual assistants, and beyond.

## 2 Problem Definition: Talking Characters

We introduce a novel task, *Talking Characters*, which focuses on generating digital characters that exhibit realistic, human-like behaviors from natural language and speech input. The task is motivated by the goal of automating dialogue-centric film production, going beyond *traditional video generation* that typically focuses on non-dialogue, narration-based scenes.

*Talking Characters* differs from conventional *talking-head* generation—which is restricted to a single face, fixed camera, and square crop—by enabling full-body character synthesis across a range of shot sizes (e.g., close-up, medium, wide) with dynamic camera motion. It supports the generation of one or more characters situated within a contextually appropriate scene.

The task is formally defined by the input-output specification and evaluation protocol, in Table 1.

| Aspect | Description |
|---|---|
| **Input** | *Text Prompt $y$*: Natural language description including (1)environment, (2)character appearance and (3)actions, (4)facing direction and emotion, (5)frame position, (6)camera movement, and (7)shot size. |
| | *Speech Audio $a \in \mathbb{R}^L$*: Raw waveform signal that drives characters' lip movements, facial expressions, and body motions. |
| **Output** | *Video $\nu \in \mathbb{R}^{T \times H \times W \times 3}$*: A rendered video of one or more talking characters (human, 3D cartoon, or animal), where $T$ is the number of frames and $(H, W)$ is the spatial resolution. |
| **Eval** | The generated characters are expected to perform well across the following five axes: |
| | *(1) Lip-Sync Quality:* Accurate and temporally aligned lip movements with respect to $\alpha$. |
| | *(2) Expression Naturalness:* Expressive and coherent facial emotions that align with both $y$ and $a$. |
| | *(3) Action Naturalness:* Realistic body gestures reflect described actions in $y$, synchronized with $a$. |
| | *(4) Text Alignment:* Coherence between visual content and $y$, including Visual layout, character appearance, and camera motion. |
| | *(5) Visual Quality:* High fidelity and temporal consistency, free from visual artifacts. |

Table 1: Task definition of Talking Characters: formalized inputs $(y, a)$, output $\nu$, and evaluation criteria.

## 3 Model: MoCha

In this section, we introduce the MoCha model, the first model to generate talking characters. We begin by outlining its architecture in subsection 3.1, followed by the localized audio attention mechanism in subsection 3.2. Next, we describe the method of generating multiple clips in subsection 3.3. Finally we provide explanation of the training strategy in subsection 3.4.

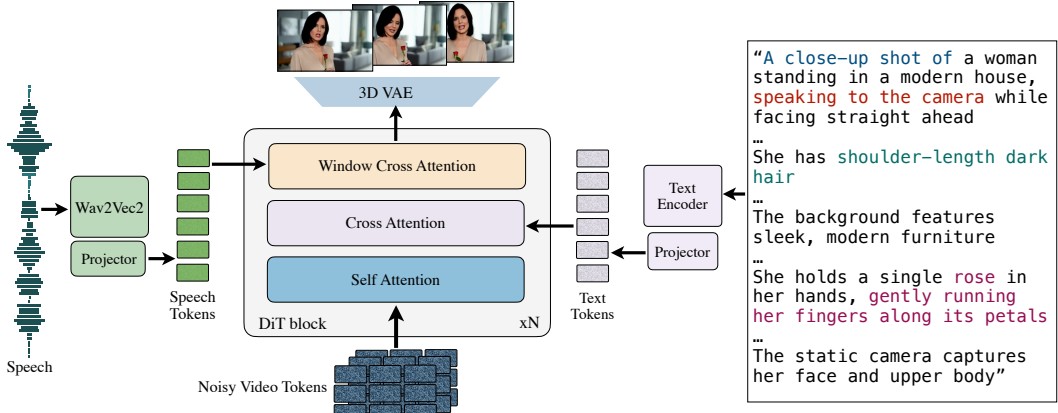

Figure 2: **MoCha Architecture.** MoCha is an end-to-end Diffusion Transformer(DiT) that generates video frames from the joint conditioning of **speech** and **text**, without relying on any auxiliary signals. Both speech and text inputs are projected into token representations and aligned with video tokens through cross-attention.

## 3.1 Audio + Text to Video Diffusion Transformers

Figure 2 presents the overall architecture of MoCha. Unlike prior works that employ text-to-image (T2I) U-Net [14, 15, 17, 22] for talking head generation, MoCha is a diffusion transformer (DiT) [23].

**Model Architecture.** MoCha adopts a fully tokenized design, where both text and speech inputs are projected into token sequences and integrated with video tokens through cross-attention. Given an video $\nu \in \mathbb{R}^{T \times H \times W \times 3}$ with $T$ frames, we encode it into a latent representation $x_0 \in \mathbb{R}^{\tau \times h \times w \times c}$ using a 3D VAE, which down-samples the video spatially and temporally. We define the temporal down-sampling ratio as $r = \frac{T}{\tau}$. Next, $x_0$ is flattened into a sequence of tokens of size $(\tau \times h \times w) \times c$ and passed to the DiT model $f_\theta(\cdot)$. Within each DiT block, the model first applies self-attention to the video tokens $x_0$, followed by sequential cross-attention with the text condition tokens $y$ and audio condition tokens $\alpha$. The audio tokens $\alpha \in \mathbb{R}^{T \times c}$ is derived from raw waveforms $a$ using Wav2Vec2 [24] and processed through MLPs to align its feature dimension with the video tokens.

**Training Objective.** We adopt Flow Matching [25], which enables efficient simulation of continuous-time dynamics, to train our model. Given a latent video representation $x_1 \in \mathbb{R}^{\tau \times h \times w \times c}$ (encoded from the input video), random noise $\epsilon \sim \mathcal{N}(0, I)$, and a continuous time step $t \in [0, 1]$, we construct an intermediate latent $x_t$ by interpolating between $\epsilon$ and $x_1$:

$$x_t = (1 - t)\,\epsilon + t\,x_1. \tag{1}$$

The model is trained to predict the velocity, defined as the difference between the data and noise:

$$v_t = \frac{dx_t}{dt} = x_1 - \epsilon. \tag{2}$$

The training loss is then:

$$L = \mathbb{E}_{\epsilon \sim \mathcal{N}(0,I),\, x_1,\, y,\, \alpha,\, t \in [0,1]} \left\| f_\theta(x_t, y, \alpha, t) - (x_1 - \epsilon) \right\|_2^2, \tag{3}$$

where $x_1$ is the latent video, $y$ and $\alpha$ are text and audio conditions, and $f_\theta(\cdot)$ is the DiT model. Unlike prior works [10, 14, 15, 20, 13, 16, 21, 17], MoCha does not rely on auxiliary objectives such as face or body masking. Instead, it learns speech-video correlations purely from data, using a fully tokenized and end-to-end training approach.

## 3.2 Localized Audio Attention

Most existing talking head generation models rely on 2D diffusion architectures (e.g., U-Net) that generate $T$ frames autoregressively, with each frame $\nu_i$ conditioned directly on its corresponding audio token $\alpha_i \in \mathbb{R}^c$. This one-to-one mapping between audio and video timesteps naturally ensures tight synchronization. In contrast, diffusion transformer (DiT) architectures depart from this design in two critical ways that complicate temporal alignment:

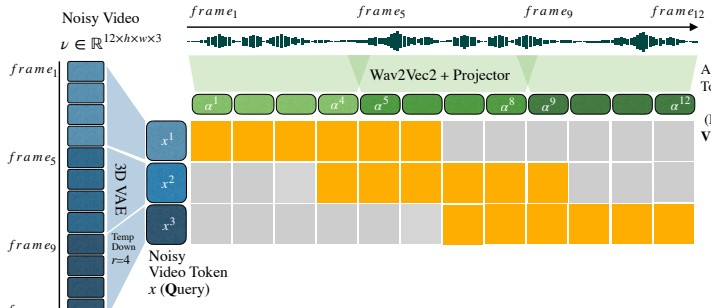

Figure 3: **MoCha's Localized Audio Attention.** To address the mismatch between compressed video tokens and high-resolution audio, MoCha employs a localized attention strategy where each video token attends only to a narrow window of nearby audio tokens. This promotes precise lip-sync and improves temporal coherence.

1. *Temporal Resolution Mismatch:* DiT-based models operate on latent representations produced by a 3D VAE, which temporally compresses videos by a factor of $r$ (commonly $r = 4$ or $8$ in recent T2V models [8, 10]). As a result, video tokens span only $\tau = T/r$ steps, while audio remains at the original resolution $T$, eliminating direct alignment.

2. *Fully Parallel Decoding:* Unlike autoregressive designs, DiT generates all $\tau$ latent frames in parallel. Without constraints, naïve cross-attention permits each video token to attend globally to the full audio sequence, which can lead to incorrect associations.

To address these challenges, we propose a *Localized Audio Attention* mechanism that introduces a temporal inductive bias. Inspired by the observation that lip movements depend on short-term phonetic patterns while body gestures and expressions reflect longer-term textual semantics, we constrain each video token's attention to a limited audio span.

Specifically, for a latent video frame $x^{(i)} \in \mathbb{R}^{h \times w \times c}$ at timestep $i \in \{1, \dots, \tau\}$, we restrict its cross-attention to audio tokens $\alpha^{(j)}$ in the window:

$$j \in [\max(1, (i-1)r-1), \ \min(T, ir+1)]. \tag{4}$$

This $r+2$-token window corresponds to the uncompressed temporal segment that $x^{(i)}$ summarizes, with one token of padding on each side for context smoothing. This simple yet effective constraint encourages alignment between speech and video while preserving local continuity across frames.

### 3.3 Multi-character Turn-based Conversation

Thanks to the clean and fully tokenized design of MoCha (see subsection 3.1), the model supports multi-clip video generation in exactly the same way as single-clip generation—**without any additional architectural modifications**. As illustrated in Figure 4, unlike video extension methods that rely on autoregressive generation conditioned on previously generated content, MoCha generates all clips in parallel. It leverages self-attention across video tokens to maintain character consistency across clips and preserve coherence in the surrounding environment.

Assuming only one character speaks at a time, we observe that speaker changes in the audio implicitly guide MoCha to transition between clips—without requiring any explicit indicators such as clip tokens [26]. Simultaneously, the text condition $y$ specifies the content of each clip.

Formally, given an audio sequence $\alpha = \{\alpha^1, \alpha^2, \dots, \alpha^T\}$, where the tokens correspond to two speakers segmented as $\alpha = [\alpha^1, \dots, \alpha^{rk}] \| [\alpha^{rk+1}, \dots, \alpha^T]$, with $r$ being the VAE temporal down-sampling ratio and $k$ being an integer index. The model generates the latent video sequence in parallel as: $x = f_\theta(y, \alpha)$, where $x = \{x^1, x^2, \dots, x^\tau\}, \tau = T/r$. The output sequence $x$ can be segmented into two clips aligned with the respective speaker turns:

$$x = [x^1, \dots, x^k] \| [x^{k+1}, \dots, x^\tau].$$

While MoCha supports seamless multi-clip generation, a challenge lies in prompting—ensuring that character attributes are consistently grounded across clips. This becomes especially difficult when characters interact or reappear in different clips. Naive captioning models typically rely on visual descriptions to refer to characters. As a result, they must repeat detailed appearance descriptions each time a character is mentioned, leading to long, redundant, and confusing prompts. For example in Figure 5. This verbosity not only increases the risk of exceeding token limits (e.g., 256 tokens) but also confuses the model during generation—especially in multi-clip scenarios.

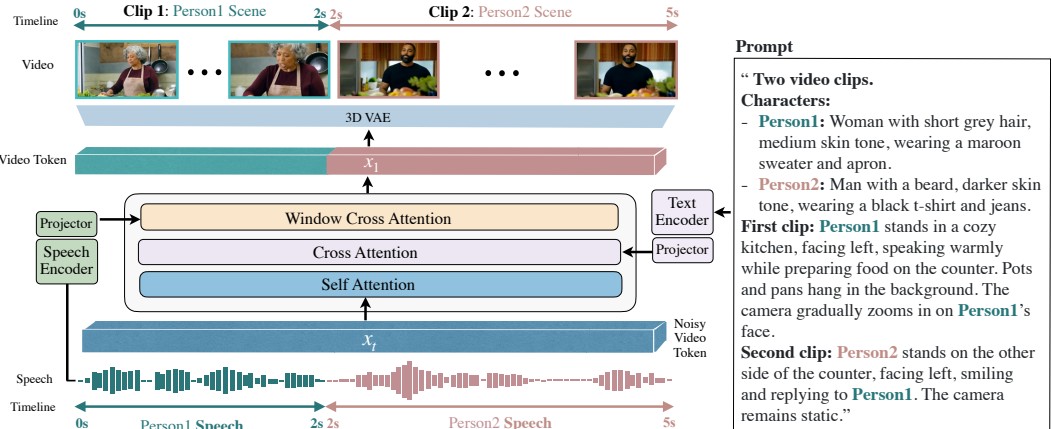

Figure 4: **Multi-character Conversation and Character Tagging.** MoCha supports generating multi-character conversion with scene cuts. We design a specialized prompt template: it first specifies the number of clips, then introduces the characters along with their descriptions and tags. Each clip is subsequently described using only the **character tags**, simplifying the prompt while preserving clarity. MoCha leverages self-attention across video tokens to ensure character and environment consistency. The audio signal implicitly guides MoCha when the transition between clips happens.

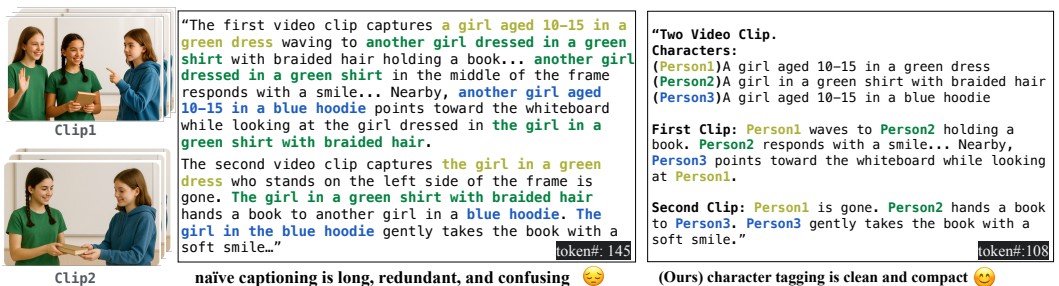

Figure 5: Character tagging provides more compact and structured prompts compared to naïve captioning.

We address this by introducing a structured prompt template with fixed keywords and a character tagging mechanism that promotes clarity, compactness, and consistency:

- **"Two Video Clip. Characters:"** Defines a list of characters, each described by visual attributes and assigned a unique tag [27] (e.g., `Person1`, `Person2`).

- **"First clip", "Second clip"** Each video segment is described using only the defined character tags.

This design significantly reduces redundancy and helps the model reliably associate visual attributes with character actions, even across multiple clips.

## 3.4 MoCha Training Strategy

We identify two key challenges in training high-quality talking character models: **(i)** *Data Scarcity and Limited Diversity:* Speech-annotated video datasets are relatively scarce and often lack sufficient visual and semantic diversity, making it difficult to directly train a MoCha model on them. **(ii)** *Varying Speech Influence Across Shot Types:* The impact of speech on video generation differs between spatial scales: it strongly governs lip and facial movements in close-up shots but has a diminishing influence in medium or wide shots involving full-body motion. Training across all shot types simultaneously may lead to slow convergence. To address both issues, we propose a **Curriculum-based Multi-modal Training strategy** that integrates modality-aware supervision with progressive visual complexity.

**Mixed-Modal Sampling.** Our data consists of a balanced mix of multimodal and unimodal data:

- 80% Multimodal (speech+text): The majority enables fine-grained audiovisual grounding.

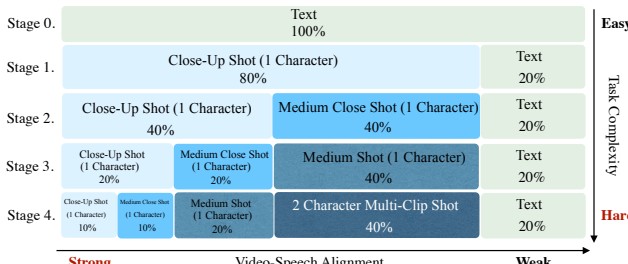

Figure 6: **Progressive Curriculum for Multimodal Training in MoCha.** MoCha is first pretrained on text-only video data to acquire general visual generation capability. The model is then progressively exposed to speech-conditioned data across different task difficulties.

- 20% Unimodal (text-only): Text-only video samples offer broader visual diversity and varied camera movements, helping the model retain strong generalization capabilities. In this setting, the speech embedding is replaced with a zero vector before the audio projector.

**Shot-Type-Based Curriculum.** We organize training into multiple stages based on shot complexity:

- Stage 0: We pretrain on large-scale text-only datasets to establish strong visual priors.
- Stages 1–N: We begin with close-up shots, which have high speech-visual correlation, and progressively incorporate more challenging scenarios such as medium/wide shots and multi-character scenes. At each stage, we halve the share of easier examples from the previous stage while maintaining the 80%/20% multimodal-unimodal ratio.

This combined strategy allows MoCha to benefit from both abundant text-only data and limited multimodal supervision while progressively mastering harder generation tasks.

## 4 Experiment

In this section, we first describe the details of our model in subsection 4.1. We then introduce MoCha-Bench for Talking Characters task and benchmark MoCha against baseline methods in subsection 4.2, and finally, we present an ablation study to analyze the impact of key design choices in Table 4.2.

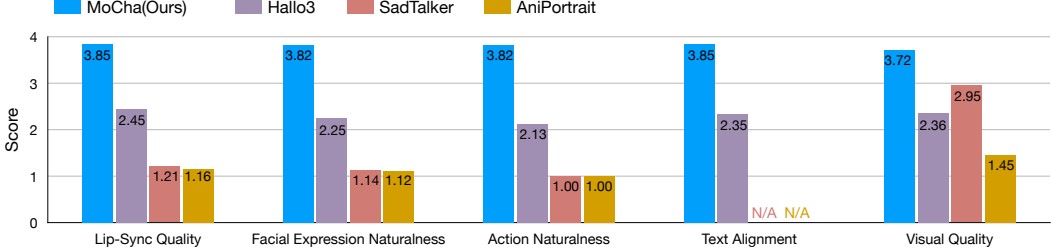

Figure 7: **Human evaluation scores with SoTA on MoCha-Bench.** Scores range from 1 to 4 across five evaluation axes, where a score of 4 reflects performance that is nearly indistinguishable from real video or cinematic production. Participants rated each method on five aspects: lip-sync quality, facial expression naturalness, action naturalness, text-prompt alignment, and visual qualit. MoCha significantly outperforms all baselines across all axes. SadTalker and AniPortrait consistently received a score of 1 for action naturalness, as these methods only perform head movements. Text alignment is marked as not applicable (N/A) for these baselines since they do not accept text input. (see appendix for table version)

### 4.1 Implementation Details

MoCha builds upon a pretrained 30B-parameter MovieGen backbone [8, 10], which we extend for speech-conditioned video generation in a fully tokenized, end-to-end setting. The model is configured to produce 128 frames at 24 frames per second, resulting in 5.3-second video clips. Training data is standardized to a spatial resolution of approximately $720 \times 720$, with flexible aspect ratios to support various shot types. To enable multimodal learning, we curate two complementary datasets: (1) a large-scale collection of $\mathcal{O}(100)$ million text-captioned videos and (2) a smaller set of $\mathcal{O}(800)$ thousand speech-annotated samples [28]. Training is distributed across 64 compute nodes. More details of the implementation and the data processing pipeline are provided in the Appendix.

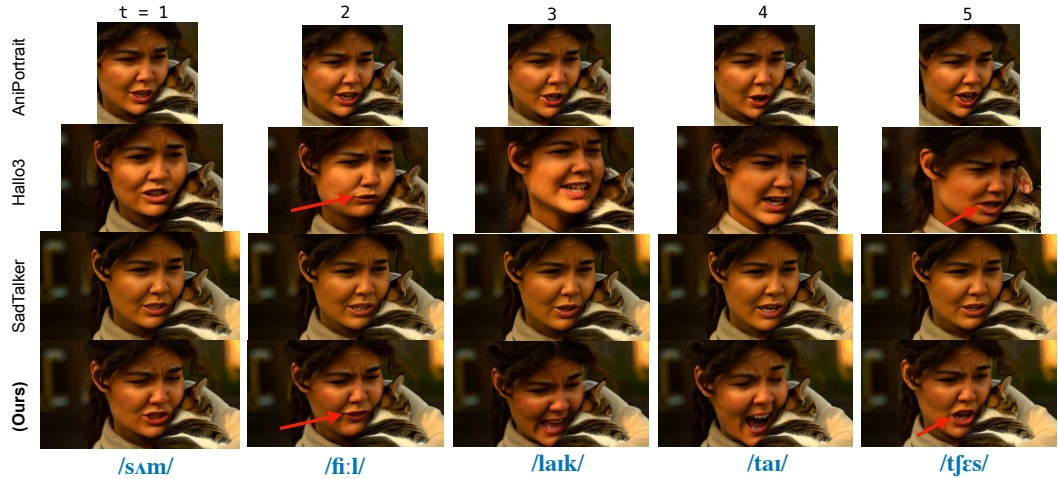

Prompt: "A close-up shot of a young woman embracing her cat outdoors. She is speaking while facing slightly to the right of the frame with an frustrating expression. The background is… she continues speaking, her expression remaining tense with anger while still facing slightly to the right…"

Figure 8: **Qualitative comparison with SoTA on MoCha-Bench.** MoCha not only produces lip movements that align closely with the input speech—enhancing the clarity and naturalness of articulation—but also generates expressive facial animations and realistic, complex actions that faithfully follow the textual prompt. In contrast, SadTalker and AniPortrait exhibit minimal head motion and limited lip synchronization. Hallo3 mostly follows the lip-syncing but suffers from inaccurate articulation, erratic head movements, and noticeable visual artifacts. Since the baselines operate in an image-to-video (I2V) setting, we provide them with the first frame generated by MoCha as input for comparison. The first frame is cropped and resized as needed to meet the requirements of each baseline.

| Method | Sync-C ↑ | Sync-D ↓ |
|---|---|---|
| SadTalker | 4.727 | 9.239 |
| AniPortrait | 1.740 | 11.383 |
| Hallo3 | 4.866 | 8.963 |
| **Ours** | **6.333** (+1.47) | **8.185** (-0.78) |

Table 2: **Comparison with SoTA on MoCha-Bench.** We report lip-sync metrics, and MoCha outperforms baselines with superior lip-sync quality.

| Ablation | Sync-C ↑ | Sync-D ↓ |
|---|---|---|
| **Ours** | **6.333** | **8.185** |
| w/o Curriculum Training | 5.659 | 8.435 |
| w/o Localized Audio Attn. | 5.103 | 8.851 |

Table 3: **Ablation on MoCha-Bench.** Removing Localized Audio Attn. degrades lip-sync, Curriculum Training improves generalization.

## 4.2 Evaluation

**Baselines.** We compare our method with state-of-the-art audio-driven talking face generation models, including SadTalker [29], AniPortrait [30], and Hallo3 [28]. These models generate talking faces conditioned on audio and auxiliary signals, such as the first frame, facial keypoints, or pose skeletons. In contrast, MoCha generates talking characters directly from raw speech and text.

**Benchmark.** We introduce **MoCha-bench**, a benchmark tailored for the Talking Character generation task. It contains 200 diverse samples, each comprising a text prompt and corresponding audio clip. The dataset spans various camera shot angles and camera movement—for example, close-up shots emphasize facial expressions and lip-sync, while medium shots highlight hand gestures and body movement. Scenes cover a wide range of human activities and object interactions (e.g., `woman holding a coffee cup`, `professor talking to student`), with characters speaking with various emotions and facing directions. All prompts were manually curated and further enriched using the publicly released LLaMA-3 [31] model to enhance expressiveness and diversity.

**Qualitative Experiments** We present qualitative results of MoCha in Figure 1, demonstrating its ability to generate diverse and realistic human motion while maintaining precise speech synchronization, even during complex actions. Contrary to the traditional view that treats audio as a weak conditioning signal requiring auxiliary supervision (e.g., pose annotations), our results show that strong audio-visual alignment can emerge purely from end-to-end training. More examples can be found in Appendix and the https://congwei1230.github.io/MoCha/.

Figure 8 illustrates a side-by-side evaluation of MoCha against baseline methods on MoCha-Bench. Since MoCha generates video directly from speech and text, while baselines operate in an image-to-video (I2V) setup, we ensure fair comparison by providing each baseline with the first frame generated by MoCha (cropped/resized to focus on the head region as needed ). Additional results are provided in the Appendix. MoCha not only produces lip movements that closely align with the speech—enhancing both articulation and naturalness—but also generates expressive facial emotion that accurately follow the textual prompt. In contrast, SadTalker and AniPortrait exhibit minimal head motion and limited lip synchronization. While Hallo3 achieves mostly consistent lip-syncing, it suffers from inaccurate articulation and erratic head movements.

**Quantitative Experiments** We evaluate video quality using the automatic metrics to measure the lip-sync quality. Table 2 presents a comparison on the MoCha-Bench. Our model achieves the best scores across all lip-sync metrics, demonstrating the effectiveness of MoCha 's end-to-end DiT design. These results further confirm that strong audio-visual alignment can emerge purely from audio conditioning without any auxiliary signal.

**Human Evaluations.** We conduct a comprehensive human evaluation to compare MoCha against baseline methods on the MoCha-Bench dataset. The evaluation is based on five axes tailored for the Talking Characters task (see section 2), with scores ranging from 1 to 4 (see Appendix). Each model output received 5 independent ratings per example, resulting in over 1000 responses per model. As illustrated in Figure 7, MoCha significantly outperforms all baselines across all five axes, with average scores approaching 4—indicating performance that is nearly indistinguishable from real video or cinematic production.

**Ablation Studies** We conduct ablation experiments to assess the individual contributions of MoCha 's core components. Table 3 presents the impact of each component. **(i)**We disable our localized audio attention mechanism during training, which results in a noticeable drop in Sync-C and increased Sync-D. We also have design variant comparison with RoPE + global audio attention in the Appendix. **(ii)**We train MoCha exclusively on speech-annotated data. This results in a noticeable drop in lip-sync quality, indicating degraded generalization due to the reduced diversity of the dataset.

# 5 Related Work

## 5.1 Talking Head Generation

Given an audio sequence and a reference face, pioneer talking-head generation works typically utilize biometric signals such as facial keypoints [32–35], or 3D priors [36–40, 29, 41] as intermediate motion representation to animate the reference face while ensuring lip synchronization. For example, SadTalker [29] first extracts 3DMM coefficients from audio and then renders the face in a 3D-aware manner. AniPortrait [30] predicts 2D facial landmarks from audio and then uses diffusion models to generate a portrait video from the 2D landmark maps. VLOGGER [42] predicts both 3D expression coefficient and 3D body pose from speech and enables the simultaneous generation of talking-face animations and upper-body gestures. Although effective, videos generated by these methods often lack expressiveness and naturalness due to the limited representation of 2D/3D priors. Recently works, such as EMO [18] and Hallo [17], generate audio-driven portrait videos end-to-end using diffusion models, which eliminate intermediate facial representations and learn natural motion from data [18, 17, 22, 16, 43]. Hallo3 [28] builds upon pretrained transformer-based video diffusion models to animate faces with dynamic head poses and background elements. Although these methods can generate natural expressions, they rely on complex auxiliary signals—such as reference images or keypoints—which not only limit the naturalness and flexibility of facial expressions and body movements but also limit the generalization ablity of those methods.

## 5.2 Video Diffusion Model

Recent diffusion-based video models have focused on improving visual quality and temporal coherence, particularly in text-guided synthesis. Early works such as Make-A-Video [44], Tune-A-Video [45], Video LDM [46], MagicVideo [47], and AnimateDiff [48] adapt text-to-image (T2I) backbones to model motion dynamics. More recent methods based on diffusion transformers—such as HunyuanVideo [10], CogVideoX [7], MovieGen [8]—along with open-source frameworks like

VideoCrafter [6], ModelScopeT2V [49], and Pyramidal Flow [50], further enhance spatio-temporal consistency and generation fidelity.

However, these models lack mechanisms for aligning speech with character behavior, often producing disjointed lip motions or gestures that fail to reflect spoken content. In contrast, MoCha pioneers a new direction by jointly conditioning on both speech and text to drive character animation—bridging the gap between realistic motion synthesis and dialogue-driven storytelling.

## 6 Conclusion

In summary, our work pioneers the task of Talking Characters Generation, pushing beyond traditional talking head synthesis to enable full-body, multi-character animations directly driven by speech and text. We present MoCha, the first framework to address this challenging task, introducing key innovations such as the localized audio attention mechanism for precise audio-visual alignment and a curriculum-based multi-modal training strategy that leverages both speech- and text-labelled data for enhanced generalization. Additionally, our structured prompt design unlocks multi-character, turn-based dialogues with contextual awareness. Comprehensive experiments and human evaluations demonstrate that MoCha delivers state-of-the-art performance in dialogue-driven video generation, marking a significant step toward scalable, cinematic AI storytelling.

## 7 Acknowledgment

We thank Xinyi Ji, Tianquan Di, Anqi Xu, Matthew Yu, Emily Luo for providing speech samples used in the MoCha demo.

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

# A    Experiments

## A.1    Audio Cross-Attention Design Ablation

To better understand how to inject audio conditioning into DiT-based video diffusion models, we conduct a systematic design study comparing multiple audio cross-attention mechanisms. Due to the high computational cost of training the full 30B MoCha model, we perform our ablation using a smaller 4B DiT backbone pretrained on MovieGen [8], and fine-tune it on 400K close-up, single-character clips with audio annotations for 200K training steps.

All models share identical training hyperparameters and differ only in the design of the audio cross-attention module. We evaluate lip-sync quality on MoCha-Bench using the SyncNet-based metrics Sync-C ($\uparrow$) and Sync-D ($\downarrow$). Results are summarized in Table 4.

**Variants.**    Below, we define the attention query $q \in \mathbb{R}^{N \times d}$ as the video tokens, and key/value $k, v \in \mathbb{R}^{T \times d}$ as the audio tokens. We denote $A(q, k, v)$ as the standard attention function.

- **Naive Audio Cross-Attention:** Each video token attends to all audio tokens without positional encoding:

$$z = A(q, k, v).$$

- **+ Learnable Positional Embedding:** Audio tokens are augmented with learnable positional embeddings $p_j$ initialized to zero:

$$z = A(q, k + p, v).$$

- **+ Sinusoidal Positional Embedding:** Instead of learned $p_j$, we add fixed sinusoidal embeddings:

$$z = A(q, k + \text{Sinusoidal}(j), v).$$

- **+ Rotary Positional Embedding (RoPE):** RoPE is applied by rotating the queries and keys:

$$z = A(\text{RoPE}(q), \text{RoPE}(k), v).$$

- **Localized Audio Attention (proposed):** Each video token $q_i$ attends only to a window of audio tokens $k_{[j_0:j_1]}$ centered on its corresponding temporal segment (see Section 3.2):

$$z_i = A(q_i, k_{[j_0:j_1]}, v_{[j_0:j_1]}),$$

  where $j_0 = \max(1, (i-1)r-1)$ and $j_1 = \min(T, ir+1)$.

- **Localized Audio Attention + RoPE:** We additionally apply RoPE to $q$ and $k$ within the window.

**Findings.**    As shown in Table 4, the naive attention baselines perform poorly in lip-sync accuracy. Adding positional information—especially sinusoidal or RoPE embeddings—significantly improves performance, suggesting that positional priors are critical for learning speech-video alignment. However, our proposed **Localized Audio Attention** consistently outperforms all other variants, demonstrating the effectiveness of constrained temporal windows for resolving the resolution mismatch between video and audio tokens.

Interestingly, adding RoPE to the Localized Attention variant slightly degrades performance, indicating potential interference between the inductive bias introduced by RoPE and the explicit temporal alignment imposed by windowing.

| Method | Sync-C ↑ | Sync-D ↓ |
|---|---|---|
| DiT-4B + Localized Audio Attention (**MoCha-4B**) | **5.692** | **8.403** |
| DiT-4B + Localized Audio Attention + RoPE | 5.027 | 9.038 |
| DiT-4B + Naive Audio Attention + RoPE | 4.872 | 8.893 |
| DiT-4B + Naive Audio Attention + Sinusoidal Embedding | 4.747 | 8.986 |
| DiT-4B + Naive Audio Attention + Learnable Embedding | 2.540 | 10.363 |
| DiT-4B + Naive Audio Attention | 2.364 | 10.385 |

Table 4: **Comparison of Audio Cross-Attention Variants on MoCha-Bench.** We report lip-sync metrics. The proposed Localized Audio Attention achieves the best performance.

## A.2 Ablation of Curriculum-Based Multimodal Training Strategy

We conduct an ablation study to assess the effectiveness of our proposed *Curriculum-based Multimodal Training* strategy (described in Section 3.4), which is designed to address two core challenges: (i) data scarcity and limited diversity in speech-annotated datasets, and (ii) varying speech relevance across spatial scales in different shot types.

To evaluate this strategy, we compare the full MoCha-30B model with two ablated variants:

- **w/o Curriculum Training:** Trained on mixed modalities (speech and text-only) but without curriculum progression (i.e., trained directly on mixed data with full shot-type complexity from the beginning).
- **w/o Mixed Multimodal Training:** Trained solely on speech-annotated videos, without any text-only data or curriculum scheduling.

All models use the same architecture and are trained for an equal number of steps.

We report both automatic lip-sync metrics (Sync-C ↑, Sync-D ↓) and three human evaluation metrics from MoCha-Bench: *Text Alignment*, *Visual Quality*, and *Action Naturalness* (detailed in Section A.4). Qualitative examples can be found on the https://congwei1230.github.io/MoCha/.

As shown in Table 5, removing the curriculum phase (*w/o Curriculum Training*) causes a moderate performance drop across all metrics, confirming the benefit of staged training that gradually increases visual complexity. Although trained for the same number of steps, the non-curriculum baseline converges more slowly and underperforms.

The *w/o Mixed Multimodal Training* variant—trained only on limited speech-annotated data—performs significantly worse, especially on *Text Alignment* and *Action Naturalness*. This confirms that unimodal speech-driven training causes overfitting to front-facing talking-face data, impairing the model's ability to generalize to diverse prompts for diverse scenes and full-body activities.

These results validate the necessity of both **modality mixing** and **curriculum progression** for robust and generalizable talking character generation.

| Method | Sync-C ↑ | Sync-D ↓ | Text Alignment ↑ | Visual Quality ↑ | Action Naturalness ↑ |
|---|---|---|---|---|---|
| **MoCha-30B (with Curriculum)** | **6.333** | **8.185** | **3.85** | **3.72** | **3.82** |
| w/o Curriculum Training | 5.659 | 8.435 | 3.17 | 3.31 | 3.27 |
| w/o Mixed Multimodal Training | 5.798 | 8.231 | 2.71 | 2.91 | 2.97 |

Table 5: **Ablation of Curriculum-Based Multimodal Training Strategy on MoCha-Bench.** We report lip-sync metrics (Sync-C ↑, Sync-D ↓) and human evaluation scores across three axes.

## A.3 Ablation of Character Tagging Strategy

We ablate the effectiveness of the *Character Tagging* strategy introduced in Section 4 for handling multi-character conversations with turn-based dialogue. This strategy is particularly important in multi-clip scenes, where multiple characters appear across different segments and speaker transitions are inferred from the audio input alone.

Our tagging mechanism assigns a unique identifier (e.g., Person1, Person2) to each character introduced at the beginning of the prompt. These tags are then reused across individual clip descriptions, allowing for clear, consistent references without repeating verbose appearance descriptions. This structured prompting significantly reduces prompt length and improves character consistency (see Figure 5).

To evaluate the impact of character tagging, we compare the full MoCha-30B model (with tagging) to a baseline trained with naïve captioning—where detailed character descriptions are repeated verbosely in each clip. Both models are evaluated on the turn-based dialogue subset of MoCha-Bench.

As shown in Table 6, removing character tagging results in a drastic drop in *Text Alignment*, indicating the model often confuses which character appears in which scene. Qualitative examples show that, without tagging, the model may generate scene-swap artifacts, fail to transition characters correctly between clips, or maintain similar lip-sync for mismatched dialogue.

We also observe degradation in *Visual Quality*, as the model occasionally blends inconsistent character features across clips. Overall, these results highlight that character tagging is essential for multi-character consistency and semantic alignment in dialogue-driven video generation.

| Method | Sync-C ↑ | Sync-D ↓ | Text Alignment ↑ | Visual Quality ↑ | Action Naturalness ↑ |
|---|---|---|---|---|---|
| **MoCha-30B (with Character Tagging)** | 4.951 | 8.601 | **3.81** | **3.64** | **3.69** |
| w/o Character Tagging | 4.465 | 8.792 | 2.01 | 2.15 | 3.03 |

Table 6: **Ablation of Character Tagging on MoCha-Bench (Turn-Based Dialogue Category).** We report lip-sync metrics (Sync-C ↑, Sync-D ↓) and human evaluation scores. Character tagging improves semantic consistency and reduces scene confusion in multi-clip dialogue scenarios.

## A.4    MoCha-Bench Human Evaluation

We conduct a comprehensive human evaluation to compare MoCha against baseline methods on the MoCha-Bench dataset. The evaluation is based on five axes tailored for the Talking Characters task , with scores ranging from 1 to 4. Each model output received 5 independent ratings per example, resulting in over 1000 responses per model. We provide the evaluation guidance as below. Besides the text guideline, we also include some visual examples to better help the annotators to judge.

This document provides evaluation guidelines, including axis definitions, scoring rubrics, and instructions for annotators. Visual examples are provided separately to support consistent judgments.

**Task Overview**

Each evaluation sample consists of:

- A generated video with audio,
- A text prompt describing the scene and character behavior.

Your task is to evaluate how well the generated video across five dimensions.

**Evaluation Axes**

- **Lip-Sync Quality:** Measures how accurately the character's lip movements align with the spoken audio.
  *Scale:* 1 – Not aligned at all, 2 – Weak alignment, 3 – Mostly aligned, 4 – Perfectly aligned.

- **Facial Expression Naturalness:** Evaluates whether facial expressions and lip-sync appear natural and contextually appropriate, without seeming robotic or exaggerated.
  *Scale:* 1 – Completely unnatural, 2 – Noticeably synthetic or stiff, 3 – Mostly natural and believable, 4 – Indistinguishable from real or cinematic performance.

- **Action Naturalness:** Assesses how naturally the character's body movements and gestures align with the audio.
  *Scale:* 1 – Completely unnatural, 2 – Noticeably unnatural, 3 – Mostly natural, 4 – Indistinguishable from real movie or TV characters.

- **Text Alignment:** Measures how well the generated actions, expressions, and presence of characters follow the behaviours described in the prompt.
  *Scale:* 1 – No alignment (e.g., missing character or major misbehavior),
  2 – Partial alignment, 3 – Mostly aligned, 4 – Perfect alignment with the prompt.
- **Visual Quality:** Evaluates the overall visual fidelity, including image sharpness, coherence, and absence of rendering issues such as artifacts, glitches, or anatomical distortions (e.g., broken limbs or unnatural body proportions).
  *Scale:* 1 – Severe artifacts, 2 – Noticeable artifacts, 3 – Mostly artifact-free, 4 – Flawless visuals.

**Annotation Instructions**

1. **Review the prompt and watch the full video at least once.**
2. **Evaluate each axis independently.** Use the provided 1–4 scale. Do not assign the same score across all axes unless truly justified.
3. **Use the full scale range.** Assign low or high scores as needed. Avoid defaulting to the midpoint.
4. **If no character is present in the video, assign a score of 1 for *Text Alignment*.**

**FAQ**

**Q: What if the same scene includes multiple characters?**
*A: Focus your evaluation on the central or speaking character as described in the prompt. Other characters may be present, but your ratings should reflect the behavior and alignment of the primary subject.*

**Q: What if the video does not include a character?**
*A: If the generated video fails to include the main character described in the prompt (e.g., an empty scene or wrong subject), you should assign a score of 1 for Text Alignment.*

| Method | Lip-Sync Quality | Facial Expression Naturalness | Action Naturalness | Text Alignment | Visual Quality |
|---|---|---|---|---|---|
| Hallo3 [17] | 2.45 | 2.25 | 2.13 | 2.35 | 2.36 |
| SadTalker [29] | 1.21 | 1.14 | 1.00 | N/A | 2.95 |
| AniPortrait [30] | 1.16 | 1.12 | 1.00 | N/A | 1.45 |
| **MoCha (Ours)** | **3.85** (+1.40) | **3.82** (+1.57) | **3.82** (+1.69) | **3.85** (+1.50) | **3.72** (+1.36) |

Table 7: **Human evaluation scores on MoCha-Bench.** Scores range from 1 to 4 across five evaluation axes, where a score of 4 reflects performance that is nearly indistinguishable from real video or cinematic production. Participants rated each method on five aspects: lip-sync quality, facial expression naturalness, action naturalness, text-prompt alignment, and visual quality. MoCha significantly outperforms prior methods across all categories. Green numbers indicate absolute improvements ($\Delta$) over the second-best method (underlined). SadTalker and AniPortrait consistently received a score of 1 for action naturalness, as these methods only perform head movements.

## A.5 MoCha-Bench Qualitative Comparison

Figure 8 presents a direct comparison between MoCha and baseline methods on MoCha. All baselines require a reference image as an auxiliary input. To ensure fairness, we first generate a video using MoCha and then use its first frame as the reference image for all baseline models. For models that do not support arbitrary aspect ratios, we crop the first frame to focus on the head region before feeding it into their networks. We provide two groups of qualitative comparisons: one featuring close-up shots and the other medium shots. The close-up group emphasizes lip-sync quality, head movement, and facial expressions, while the medium shot group focuses on hand movements during speech. MoCha not only produces lip movements that closely align with the input speech—enhancing both articulation and naturalness—but also generates expressive facial animations and realistic, coordinated actions that accurately follow the textual prompt. In contrast, SadTalker and AniPortrait exhibit minimal

head motion and limited lip synchronization. While Hallo3 achieves mostly consistent lip-syncing, it suffers from inaccurate articulation and erratic head movements. In the medium shot comparisons, Hallo3 also introduces noticeable visual artifacts, particularly during complex actions.

# B  MoCha Model Architecture

## B.1  3D VAE

Our 3D-VAE is based on a variational autoencoder and compresses the input pixel space video $\nu$ of shape $\in \mathbb{R}^{T \times H \times W \times 3}$ into a lower-dimensional, continuous latent representation $x_0$ of shape $\tau \times h \times w \times c$, . In our implementation, we compress the input $8\times$ across the spatial dimension while $4\times$ across the temporal dimension, i.e., $H/h = W/w = 8$ and $T/\tau = 4$ The latent channel dimensionality is fixed at $c = 16$ for all experiments reported herein.

## B.2  Text Encoder

We employ a triad of text encoding architectures—UL2, ByT5, and a Long-prompt variant of MetaCLIP—to equip the backbone with both high-level semantic and fine-grained character-level textual comprehension. Each encoder generates a sequence of 256 token embeddings. To unify these heterogeneous representations, we apply dedicated linear projection layers and LayerNorm to each encoder's output, transforming them into the model dimension 6144-dimensional feature space. The resulting normalized embeddings from all three streams are then concatenated to produce the final comprehensive text representation fed into the backbone. Among these, the MetaCLIP encoder specializes in generating text features inherently aligned with visual modalities, optimizing performance for cross-modal generation tasks. UL2, conversely, excels at encoding deep linguistic reasoning and semantics, While ByT5 captures character-level details, making it effective for encoding visual text.

## B.3  Audio Encoder

Our audio pipeline is powered by Wav2Vec2, but instead of relying solely on its final output, we extract and stitch together the embeddings from all 12 internal layers. This approach gives us a deeper, layered view of the audio content, with each layer contributing a 768-dimensional slice to the overall representation. After running the audio through Wav2Ve2's tokenizer, we stretch or compress the resulting sequence using linear interpolation before Before the audio hits Wav2Vec2. So that we end up with the same number of audio features as there are video frames—effectively assigning a unique audio token to each frame. To provide each frame's audio token with extra context, we expand its feature vector by gluing on the tokens from the five frames before and after it. So for any given frame $f$, the final embedding is built as $A(f) = [A(f - 5), \ldots, A(f), \ldots, A(f + 5)]$. This chunky, context-aware audio feature then passes through a straightforward two-layer neural net (an MLP with a hidden size of 512) which reshapes it into the 6144-dimensional token $\alpha(f)$ needed by our model's backbone.

## B.4  DiT Architecture

The core architectural hyperparameters of our MoCha-30B DiT backbone are provided in Table 8.

**Localized Audio Cross-Attention.**    To incorporate temporal locality in the audio stream, we adopt a windowed cross-attention mechanism with a window size of $r + 2 = 6$, where $r = 4$ is the temporal downsampling factor of the video encoder. For a latent video frame $x^{(i)} \in \mathbb{R}^{h \times w \times c}$ at timestep $i \in \{1, \ldots, \tau\}$, attention is restricted to audio tokens $\boldsymbol{\alpha}^{(j)}$ falling within the interval:

$$j \in [\max(1, (i - 1)r - 1), \ \min(T, ir + 1)], \tag{5}$$

which corresponds to the original (pre-downsampled) temporal span of $x^{(i)}$ with one token of padding on each side to enable smoother context transitions at boundaries.

**Modality Integration Pathway.**    Each attention block integrates modalities in a sequential manner. It begins with video self-attention, followed by localized audio cross-attention and then text cross-

| Layers | Model Dimension | FFN Dimension | Attention Heads | Activation Function | Normalization |
|--------|-----------------|---------------|-----------------|---------------------|---------------|
| 48 | 6144 | 16384 | 48 | SwiGLU | RMSNorm |

Table 8: Core architecture hyperparameters for the MoCha-30B Transformer. The model has 30 billion parameters in the Transformer stack alone, excluding auxiliary components such as text embedding models, speech embedding models, and the 3D-VAE.

attention. Each stage incorporates a residual connection, modulated by a scalar weight. In our implementation, these residual weights are fixed to 1. The full process is given by:

$$(1) \quad \mathbf{z}_{\text{video}}^{\text{out}} = \text{SelfAttn}_{\text{video}}(\mathbf{z}_{\text{in}})$$

$$(2) \quad \mathbf{z}_{\text{text}}^{\text{out}} = \text{CrossAttn}_{\text{text}}(\mathbf{z}_{\text{video}}^{\text{out}}, \boldsymbol{y}) + \lambda_{\text{text}} \cdot \mathbf{z}_{\text{video}}^{\text{out}}$$

$$(3) \quad \mathbf{z}_{\text{audio}}^{\text{out}} = \text{CrossAttn}_{\text{audio}}(\mathbf{z}_{\text{text}}^{\text{out}}, \boldsymbol{\alpha}) + \lambda_{\text{audio}} \cdot \mathbf{z}_{\text{text}}^{\text{out}}$$

where:

- $\mathbf{z}_{\text{in}}$ is the input latent sequence.
- $\text{SelfAttn}_{\text{video}}$ denotes self-attention over video tokens.
- $\text{CrossAttn}_{\text{audio}}(\cdot, \boldsymbol{\alpha})$ performs localized cross-attention with audio tokens $\boldsymbol{\alpha}$.
- $\text{CrossAttn}_{\text{text}}(\cdot, \boldsymbol{y})$ performs cross-attention with global text tokens $\boldsymbol{y}$.
- $\lambda_{\text{audio}}$ and $\lambda_{\text{text}}$ are residual weights, both set to 1 in our implementation.

This staged fusion enables progressive enrichment of the video representation by incorporating auditory and textual information, while preserving intermediate features through residual addition.

## C  MoCha Training

We provide the training details for our MoCha-30B model in Table 9. We used a constant learning rate scheduler with 2000 warm-up steps. We use a Progressive Curriculum for Multimodal Training as describe in subsection 3.4.

**Mixed-Modal Sampling.** Our data consists of a balanced mix of multimodal and unimodal data:

- 80% Multimodal (speech+text): The majority enables fine-grained audiovisual grounding.
- 20% Unimodal (text-only): Text-only video samples offer broader visual diversity and varied camera movements, helping the model retain strong generalization capabilities. In this setting, the speech embedding is replaced with a zero vector before the audio projector.

**Shot-Type-Based Curriculum.** We organize training into multiple stages based on shot complexity:

- Stage 0: We pretrain on large-scale text-only datasets to establish strong visual priors.
- Stages 1–N: We begin with close-up shots, which have high speech-visual correlation, and progressively incorporate more challenging scenarios such as medium/wide shots and multi-character scenes. At each stage, we halve the share of easier examples from the previous stage while maintaining the 80%/20% multimodal-unimodal ratio.

This combined strategy allows MoCha to benefit from both abundant text-only data and limited multimodal supervision while progressively mastering harder generation tasks.

We build MoCha based on Movie Gen Backbone. The Stage 0 training is included in the Movie Gen backbone pertaining. Then we add the speech cross attention and speech projector to the Movie Gen backbone to build MoCha. During Stages 1-N training, We full-finetuning the entire 30B MoCha model while freezing the text encoder and speech encoder and text projector. Throughout training, all

input examples are resized to a resolution of approximately 720 px, preserving their original aspect ratios.

| Stage | #GPUs | Global bs | LR | Shot Types | #Iters | #Audio Samples | #Text Samples |
|-------|-------|-----------|-----|------------|--------|----------------|---------------|
| 0 | 1024 | 512 | 1e-5 | Images / Videos | 200K | 100M | – |
| 1 | 512 | 512 | 1e-5 | Close-Up (1 Char) | 200K | 400k | 100M |
| 2 | 512 | 512 | 1e-5 | Close-Up / Medium Close (1 Char) | 200K | 300k | 100M |
| 3 | 512 | 512 | 1e-5 | Close-Up / Medium / Medium Close (1 Char) | 100K | 200k | 100M |
| 4 | 512 | 512 | 1e-5 | Close / Medium / Multi-Char / Multi-Clip | 100K | 100k | 100M |

Table 9: Progressive Curriculum for Multimodal Training in MoCha. Stages gradually increase in temporal and compositional complexity, progressing from static images and short clips to multi-shot, multi-character, dialogue-driven video. Residual text supervision is maintained throughout.

# D   Data Processing Pipeline

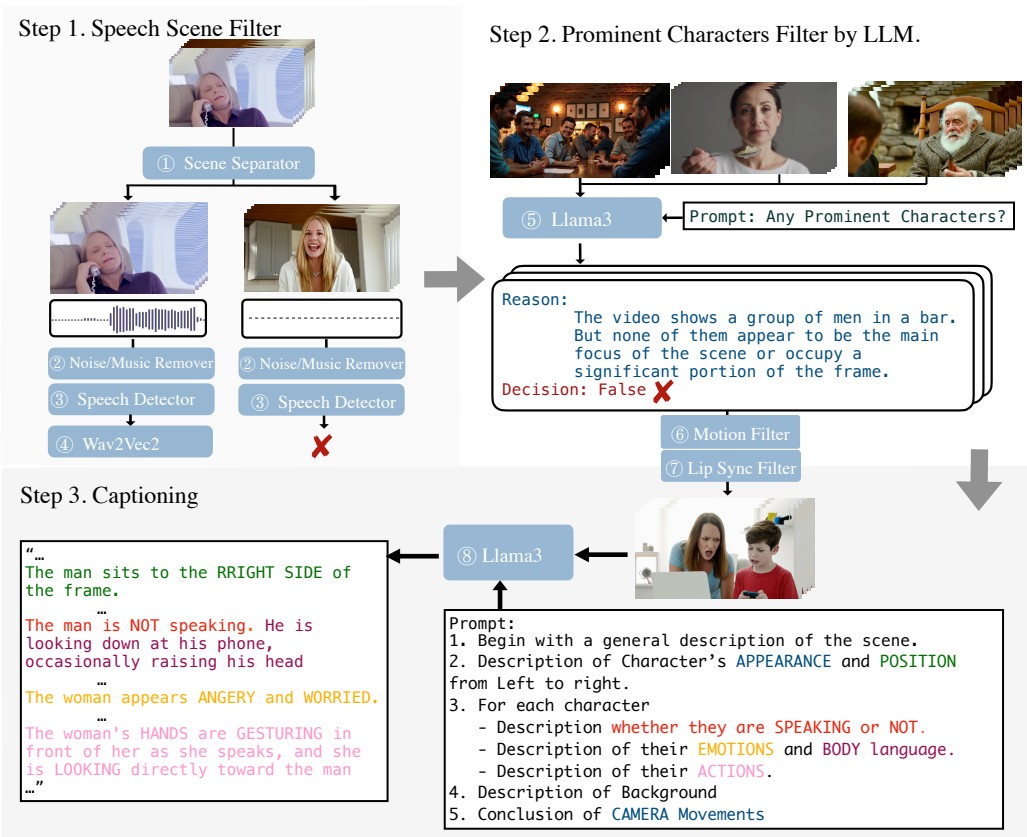

Figure 9: **Data Processing Pipeline.** Our four-stage pipeline includes: (1) **Speech Scene Filtering**, (2) **Prominent Character Filtering**, (3) **Motion and Lip-Sync Verification**, and (4) **Scene Captioning**. Together, these steps produce high-quality, speech-aligned training samples.

To ensure high-quality supervision for speech-conditioned video generation, we develop a four-stage data processing pipeline, as illustrated in Figure 9. Each stage is designed to filter noisy data and produce rich, structured annotations for training.

- **(1) Speech Scene Filtering:** Raw videos are first segmented into scenes using PySceneDetect [51]. We then detect and retain segments containing clean, spoken audio by removing clips with excessive background noise or dominant music. For valid segments, we apply noise suppression and extract speech embeddings using Wav2Vec2 [52, 24].

- **(2) Prominent Character Filtering:** To focus on scenes with clearly visible speakers, we use an LLM-based filter that analyzes visual cues and removes clips lacking a central character figure. This step emphasizes narrative relevance and visual clarity.
- **(3) Motion and Lip-Sync Verification:** We further refine the data by checking for facial and body motion aligned with speech. Segments without meaningful movement or with weak audio-visual correspondence are excluded.
- **(4) Scene Captioning:** Finally, a large language model [53] is used to generate structured captions for each clip. These include detailed descriptions of character appearance, spatial layout, speaking behavior, emotional expression, and physical gestures—enabling rich conditioning during training.

This pipeline results in a curated dataset of approximately 300 hours of high-quality, speech-aligned video content, totaling around 800K annotated samples.

# E   Limitations

Despite the strong performance of our model across various talking character scenarios, we identify several limitations that affect generation quality under certain conditions. We provide corresponding examples on the https://congwei1230.github.io/MoCha/.

*Failure to Lip Sync in Wide or Extreme Wide Shots:* When the input caption is too vague—particularly lacking details about facial attributes or shot type—the model may default to generating a wide or extreme wide shot. In such cases, the character often appears too far from the camera, and the lip region contains only a few pixels. As a result, the model may fail to perform accurate lip synchronization. *Example prompt: "A man playing skateboard at a skatepark."*

*Multiple Characters in Scene:* While the model is generally capable of making the intended character speak in scenes with multiple characters, we observed occasional confusion about which character should be speaking once both characters' face are visible in a single shot. This can lead to degraded lip-sync quality. The limitation likely stems from a scarcity of similar multi-character examples in the training data, where two or more characters appear in the same shot and speak to the camera. *Example prompt: "A medium shot set in a dimly lit tavern. The central figure, a rugged man with long, wet-looking hair and a thick beard, sits on a rustic wooden bench. He wears a weathered wool cloak over leather armor, evoking the image of a battle-hardened warrior. His expression is intense as he speaks, holding a short sword in his right hand. To his left, another man with tied-back hair and fur-lined garments watches him closely."*

*Over-Expression with High Speech CFG:* Increasing the speech classifier-free guidance (CFG) value beyond the default (e.g., from 7.5 to 12) can cause the model to generate characters with overly expressive facial and body motions. While this can improve speech emphasis, it may also reduce realism or break immersion in otherwise grounded scenes. *Example prompt: "A tracking shot circling around the man as he ties a tie over his blue suit. He speaks to the camera while adjusting the knot, maintaining eye contact throughout."*

# F   Broader Impact

The goal of MoCha is to advance the field of dialogue-centric video generation and enable creative professionals—such as filmmakers, animators, educators, and content creators—to produce emotionally engaging character-driven videos using natural language and speech inputs. By lowering the barrier to cinematic-quality storytelling, this technology democratizes digital media production and unlocks new possibilities for interactive entertainment, educational content, and virtual communication.

However, as with any powerful generative technology, there are important societal and ethical considerations to address.

**Misuse and Synthetic Media Risks.** The use of speech-driven character generation raises concerns around the potential misuse of synthetic media. While MoCha synthesizes characters and scenes entirely from noise, without cloning real human identities or voices, there is still a risk that the generated videos could be misrepresented as real footage—particularly if aligned with real-world audio. This raises issues of misinformation, media authenticity, and potential psychological manipulation.

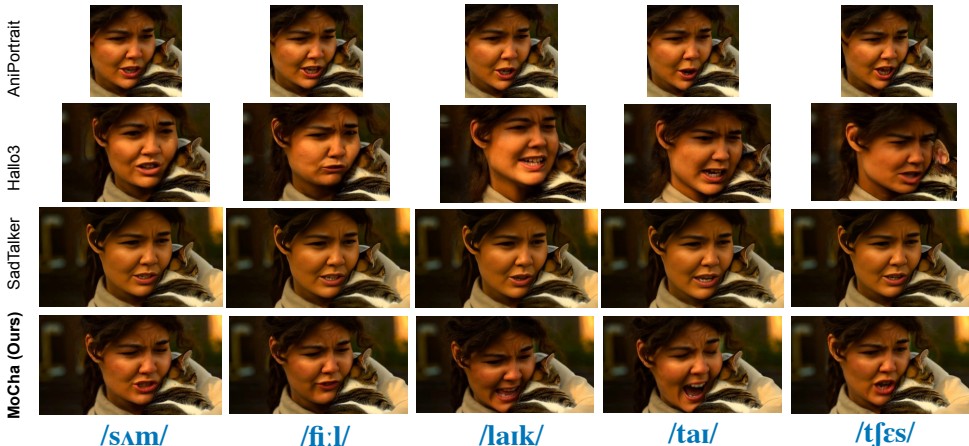

Prompt: "A close-up shot of a young woman embracing her cat outdoors. She is speaking while facing slightly to the right of the frame with an frustrating expression. The background is… she continues speaking, her expression remaining tense with anger while still facing slightly to the right…"

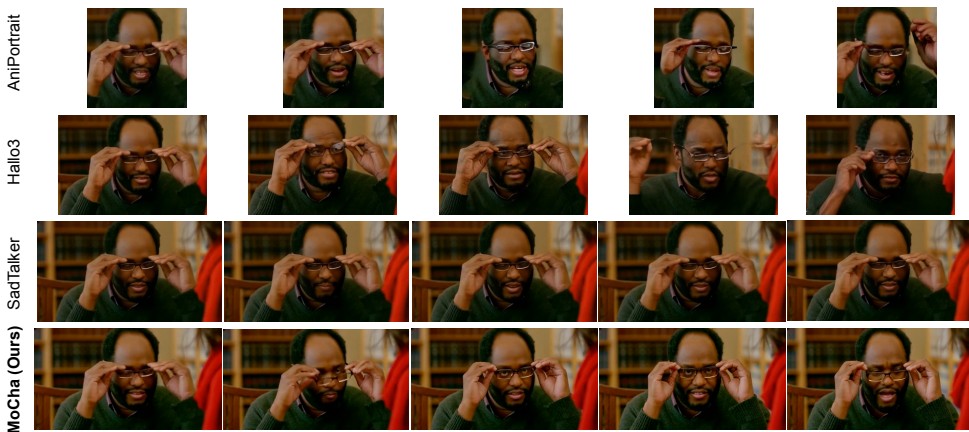

Prompt: "A medium shot of a man speaking while adjusting his glasses… As he talks, he first takes off his glasses, then puts them back on…"

Figure 10: **Qualitative comparison between MoCha and baselines on MoCha-Bench.** MoCha not only produces lip movements that align closely with the input speech—enhancing the clarity and naturalness of articulation—but also generates expressive facial animations and realistic, complex actions that faithfully follow the textual prompt. In contrast, SadTalker and AniPortrait exhibit minimal head motion and limited lip synchronization. Hallo3 mostly follows the lip-syncing but suffers from inaccurate articulation, erratic head movements, and noticeable visual artifacts. Since the baselines operate in an image-to-video (I2V) setting, we provide them with the first frame generated by MoCha as input for comparison. The first frame is cropped and resized as needed to meet the requirements of each baseline.

Unlike deepfake systems that typically manipulate real people's faces, MoCha does not operate on real identity inputs. All visual content is generated from scratch using text and speech prompts. Nevertheless, the audio guidance—if paired with sensitive or impersonated speech—could still be used to create misleading portrayals. To mitigate this risk, we recommend responsible disclosure practices and support for watermarking or synthetic media detection tools.

**Privacy and Consent.** Although MoCha does not require real human images or voices for generation, broader deployment of similar technologies in the future may prompt privacy concerns, especially if adapted to personalized avatars or voice-based likenesses. To ensure responsible use, it is critical to uphold strict data privacy standards, including transparent data usage policies, informed consent when training on human likenesses, and tools for individuals to opt out of potential misuse.

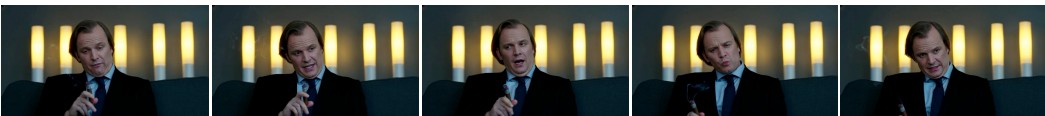

Prompt: "A close-up shot of a man sitting on a dark gray couch… Behind the man are three white cylindrical light fixtures with yellow lights inside them… the man continues to speak to the camera while he **holds a lit cigar, the smoke curling gently into the air**…"

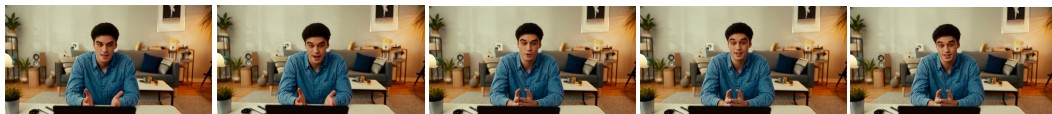

Prompt: "A medium shot of a young man aged 25 to 35 is sitting in the living room in a leisurely environment… He is live-streaming, sitting in front of a desk with a laptop in front of him. His demeanor is relaxed and friendly, **gesturing with his hands while speaking**…"

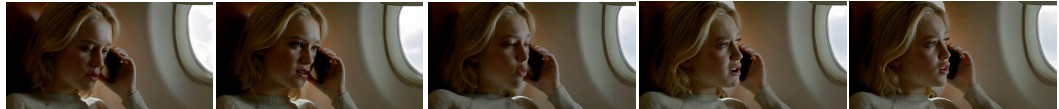

Prompt: "A close-up shot of a young blonde woman sitting in an airplane seat, facing slightly to the right as she talks on the phone with a **worried expression.** As the video progresses, she continues speaking and **eventually turns to look out the window**…"

Figure 11: **Qualitative results of MoCha on MoCha-Bench**. MoCha not only generates lip movements that are well-synchronized with the input speech, but also produces natural facial expressions that reflect the prompt along with realistic hand gestures and action movements

**Bias and Representation.** As with many generative models, outputs from MoCha may reflect biases present in the underlying training data—particularly in terms of character appearance, behavior, or cultural representation. Careful dataset curation and evaluation are needed to ensure diversity and inclusiveness, and to avoid reinforcing harmful stereotypes or excluding underrepresented groups in generated media.

**Ethical Deployment.** We encourage deployment of MoCha only in contexts that respect consent, truthfulness, and creative integrity. As the technology matures, we advocate for the establishment of community-driven ethical guidelines, collaboration with media regulators, and public education on the capabilities and limits of AI-generated video.

By proactively identifying and addressing these risks, we aim to support the safe and beneficial advancement of talking character technologies and ensure they are developed and applied in ways that are ethical, inclusive, and socially responsible.

