# OpenReview forum: "MoCha: Towards Movie-Grade Talking Character Generation"
_NeurIPS.cc/2025/Conference — NeurIPS 2025 spotlight_

### Official Review · Reviewer_qDos · 2025-06-29

**Clarity:** 3
**Significance:** 2
**Originality:** 2
**Rating:** 4
**Confidence:** 4

**Summary:**

This paper proposes a text-to-video model that supports speech audio conditioning. It details the implementation of progressive training and audio conditioning techniques. Built on a 30B base model, the approach outperforms baseline methods across multiple evaluation metrics.

**Questions:**

- Using audio conditioning with features extracted from adjacent frames and multi-stage training is very common in audio-conditioned video generation methods such as EMO, Hallo, Hallo3, and OmniHuman. A clearer explanation of the paper’s unique contributions in this context would be helpful.
- The current experimental comparison setup is inconsistent for other methods, making fair evaluation difficult, as noted in the weaknesses. It is recommended that the authors compare their method directly with other text-to-video approaches (e.g., using the metrics in Figure 7) while excluding audio-related metrics, which are already clearly reported in Table 2.
- The authors should provide some evaluation or discussion of the base model’s performance itself, as this would improve the completeness and clarity of the paper.

**Ethical Concerns:**

["NO or VERY MINOR ethics concerns only"]

**Final Justification:**

This paper demonstrates that multi-character, turn-based conversational video generation can be achieved through joint data training and specific architectural designs. The results are impressive. However, in the initial submission, the description of certain methods lacks clarity, particularly the audio conditioning method. The authors will revise these descriptions in the final version. I recommend this paper for acceptance.

**Limitations:**

- The paper has some questionable experimental settings, such as comparing image-to-video methods using images generated by the proposed text-to-video model.
- It lacks analysis and comparison with other text-to-video models, including the base video generation model.
- The method’s contributions are unclear, as audio conditioning and multi-stage training are common practices in this field.

**Paper Formatting Concerns:**

No concerns regarding paper formatting.

**Quality:**

3

**Strengths And Weaknesses:**

Strengths:
1. The paper is clearly written, providing detailed explanations of the implementation.
2. The visual results are strong, showing noticeable improvements over baseline methods.

Weaknesses:
1. The experimental setup is flawed: the baselines compared are image-plus-audio to video models, but the test images come from the proposed text-to-video model. This likely makes the test images out-of-domain for the baselines but in-domain for the proposed method, resulting in an unfair comparison.
2. The method shares many similarities with existing work, such as the audio conditioning approach and multi-stage training, but the paper lacks clear discussion or analysis of these overlaps.
3. The method relies on a large, strong base video generation model which differs from other methods and this difference is not addressed or analyzed in the paper.

---

> ### Author Rebuttal · Authors · 2025-07-31
>
> Dear reviewer `qDos`,
>
> We sincerely appreciate the time and effort you have dedicated to reviewing our work and providing valuable feedback. Below, we are pleased to share our detailed responses.
>
> **W1&L1 Use the test images from the MoCha are likely out-of-domain for the baselines. Q2 & Q3 &L2: Recommended to compare MoCha directly with other t2v approaches and the base model**
>
> **Answer:** We thank the reviewer for raising this important point!  Our work is dedicated to the novel task: **talking characters**.
> The experiments were designed to compare with existing methods that can only do **talking head** generation.
> To follow their inference settings, we used the first frame from MoCha as input.
> While we cropped the frames to match baselines' training distribution and avoid blurry or artifacts, we acknowledge that potential distribution mismatch may still exist.
> We conducted further experiments using three diverse groups of inputs to test the robustness of baselines.
>
> Group 1: 3–5 real images from the Internet matching the prompts (e.g., “woman driving a car”), with clear head poses and scene alignment.
>
> Group 2: 5 synthetic images generated by CogView4, GPT-Image-1, and FLUX.1-dev.
>
> Group 3: MoCha-generated first frame.
>
> **Key Observations: (1)Baseline performances are robust across groups, showing MoCha-generated frames are not out-of-domain for them. (2)MoCha significantly outperforms all baselines, confirming its superiority is not due to biased inputs**.
>
> In response to Q2 & Q3 &L2.  We thank the reviewer for the suggestions! and compared MoCha with other t2v models as required.  We observed that since mocha is trained on talking human videos, it achieves the best facial expression realism. It can handle complex prompts like "speaking while gently sipping water." while other models fail to produce talking behavior in this case. This comparison further supports our claim:
> **MoCha bridges the gap between Movie Generation and Talking-Head Synthesis, making cinematic storytelling possible.**
>
> | Method|Sync-C↑| Sync-D ↓| Facial Expression | Action Naturalness | Text Alignment | Visual Quality |
> |-|-|-|-|-|-|-|
> | Hallo3 (Group 1)| 4.820 | 8.899   | 2.33  | 2.33| 2.35  | 2.32|
> | Hallo3 (Group 2)| 5.024 | 9.263   | 2.38  | 2.35| 2.38 | 2.36|
> | Hallo3 (Group 3)| 4.866 | 8.963   | 2.25  | 2.13| 2.35 | 2.36 |
> | SadTalker (Group 1)   | 4.824 | 9.133   | 1.35   | 1.00 | N/A| 2.98       |
> | SadTalker (Group 2)   | 4.569 | 9.310   | 1.04    | 1.00 | N/A | 2.93       |
> | SadTalker (Group 3)   | 4.727 | 9.239   | 1.14 | 1.00      | N/A            | 2.95       |
> | AniPortrait (Group 1) | 1.749 | 11.005  | 1.17 | 1.00   | N/A| 1.46       |
> | AniPortrait (Group 2) | 1.688 | 12.084  | 1.13  | 1.00      | N/A | 1.45       |
> | AniPortrait (Group 3) | 1.740 | 11.383  | 1.12| 1.00      | N/A            | 1.45       |
> | Kling1.6 | N/A   | N/A     | 3.51    | 3.12      | 3.15          | 3.74       |
> | Wan2.1  | N/A   | N/A     | 3.48    | 3.79      | 3.48          | 3.67       |
> | OpenAI SoRA  | N/A   | N/A     | 3.25  | **3.83**  | 3.21   | **3.89**   |
> | MoCha(base T2V model)| N/A   | N/A     | 3.49              | 3.82      | 3.16          | 3.73       |
> | MoCha   |**6.037**|**8.103** | **3.82** | 3.82      | **3.85**      | 3.72       |
>
>
> **W2 & Q1 & L3: Lack of Discussion on the uniqueness of audio conditioning and multi-stage training strategy**
>
> We provide discussions below, followed by supporting results.
>
> **1. Localized audio attention**
> Most talking head generation models (EMO, Hallo) employ 2D Unet model and align audio and video features on a per-frame basis. DiT utilizes 3D VAEs with temporal compression of video frames, creating a resolution mismatch between audio and video features. Notably, no existing work (Hallo3, omnihuman) thoroughly studies how to align audio and video within a DiT framework effectively. In Appendix Section A.1, we are among the first to conduct a comprehensive ablation on this, evaluating (1) Naive Audio Cross Attn (2) Naive Cross-Attn with positional embeddings (RoPE, learnable, sinusoidal) (3) Our proposed localized audio attention. Our method consistently outperforms all variants. This study offers valuable insight for future research. Furthermore, we compare it to Hallo3’s alignment strategy. Hallo3 temporally compresses audio features twice via stacked convolutions to match the video tokens. We find that this compression degrades fine-grained audio details, making it challenging to render characters accurately pronouncing phonemes like /iː/, /ai/, /tʃ/, and /θ/. In contrast, our method avoids temporal compression. We constrain each video token to only attend to a local window of audio tokens, which preserves the fine-grained details of the audio signal.
>
> We provide a direct comparison of both methods on a DiT-T2V-4B backbone. Our approach results in superior audio-visual synchronization.
> |Method |Sync-C↑| Sync-D ↓|
> |- |-  | - |
> |DiT-4B + Localized Audio Attention (MoCha-4B)|  5.692 |8.403|
> |DiT-4B + Hallo3 Audio Attention|  4.885 |8.896|
>
>
> **2. Training Strategy**
> Our training strategy contains two parts **Shot-Type-Based Curriculum** + **Mixed-Modal Sampling**.
>
> Existing works(hallo, hallo3, omnihuman) do use multi-stage training. However, they use it to add auxiliary signals, gradually introducing them (face image → audio → pose) to the model. At the audio stage, they simply feed all audio-annotated data into the model. This works fine as these models' tasks are talking heads generation, and the data are mostly cropped human face videos with similar face sizes. But for MoCha, our task and data are much more complex, involving talking faces→half-body gesture→full-body movements→camera control→multi-character, multi-clip.  As the visual scope increases, the mouth region becomes smaller and harder to supervise. Training on all data types simultaneously to learn audio control is clearly inefficient, especially since MoCha doesn't use strong control signals like reference frames. To tackle this, our **Shot-Type-Based Curriculum** is not used to add new signals like existing work, but instead structures the learning process based on shot complexity. **This method significantly improves convergence, as shown in Appendix Table 5.**
>
>
> SoTA methods(including hallo3, omnihuman) struggle with generating characters that go beyond static poses and fixed camera views. They fail to generalize to realistic cinematic scenarios, such as characters talking while expressing emotions, naturally performing actions, interacting with objects, or moving alongside the camera in response to text conditions. We identify two reasons (1) They overlook the richness of text conditions, often reducing prompts to oversimplified inputs like “a person is talking.” (2)They rely exclusively on human data, sampled primarily from audio/pose-driven datasets. To directly address this, we propose **Mixed-Modal Sampling**: We always provided high-quality and multi-grained captions to the model (as shown in Figure 2); During training we consistently sample 20/% of the data from a diverse text2video dataset.
>
> **This method enables MoCha to achieve superior generalization** in scenarios involving complex lighting, human-animal interactions, and unlock the universal controllability via prompts eventually **unlock the multi-character turned based converstation ablity**.
>
> We provide the following experiment against the strongest baseline Hallo3.
> The base models of Hallo3 and MoCha-4b have similar size and on-pair performance initially.
> After learning the audio condition, MoCha-4b outperforms Hallo3 on all axes. Hallo3 suffers from worse generalization, as evidenced by its degraded performance in text alignment and action naturalness compared to its base model.
> This underscores the uniqueness of our contribution.
> | Method |Sync-C↑| Sync-D ↓| Facial Expression | Action Naturalness | Text Alignment | Visual Quality |
> |-  | -  | - |-  |- |- | - |
> | CogVideoX-5b (Base for Hallo3) | N/A   | N/A     | 2.88  |  2.85    | 2.91| **2.83**       |
> | Hallo3  | 4.866 | 8.963   | 2.25 (-0.63)      |  2.13 (-0.72)      | 2.35 (-0.56)   | 2.36 (-0.47)   |
> | T2V Base for MoCha-4b    | N/A   | N/A     | 2.86    |  2.76   | 2.93   | 2.71|
> | MoCha-4b  | **5.692** | **8.403**  | **2.92** (+0.06)  | **2.87** (+0.11)   | **2.94**(+0.01)| 2.69 (−0.02)   |
>
> **W3: The method uses a large, strong base video generation model differs from the others**
>
> As shown in the tables above, our proposed methods consistently outperform baselines, even when using base models of similar size and capability. This demonstrates the effectiveness of our approach, not limited to large models.
> Our research goal—as stated in the claims—is to **bridge the gap between generic video generation and constrained talking-head synthesis**. We also want to deliver the message that **movie-grade storytelling can be achieved without relying on auxiliary prior signals**.
> However, as shown in the table above, base models with 4B/5B parameters still fail to consistently generate videos with a score of 3, meaning they cannot reliably produce mostly artifact-free and prompt-following results.
> Therefore, to support our goal, we intentionally build on top of a large base model to ensure satisfactory visual fidelity in the results.
>
> **L3: The method’s contributions are unclear.**
>
> We would like to clarify and highlight the key contributions of our work:
> MoCha is **among the first to bridge the gap between video generation and talking-head synthesis, making cinematic storytelling possible**. We introduced a new task —talking characters, along with a corresponding benchmark and a promising model for this task.
> MoCha is also  **among the first to demonstrate that movie-grade storytelling can be achieved without relying on auxiliary prior signals**. Finally, it is **the first to support multi-character, turn-based dialogue clips generation.** A capability not achieved by any existing method to date.

---

> > ### Comment · Reviewer_qDos · 2025-08-03
> >
> > Thank you for the detailed response, which resolved most of my questions. I hope we can continue the discussion, as I have one remaining point regarding the novelty of the method.
> >
> > First, on the multi-stage training. I understand the logic of progressively enriching features. As a point of discussion, while some current methods like Hallo3 (indicated in Figure 1) also use relatively complex captions, I recognize that the authors' training setup was specifically designed to enable their proposed features. Given the impressive results, I agree that this is a fair approach.
> >
> > Second, concerning the Localized Audio Attention, I would appreciate some further clarification on its novelty. Having re-examined the Naive Audio Cross-Attention variants in the supplementary material, my understanding is that having all video tokens attend to all audio tokens is a relatively uncommon approach. Since early diffusion-based methods like EMO, the mainstream practice has been for each frame to attend to audio tokens near its timestamp. This context helps explain the improvement over the other variants, indeed, this outcome feels like what one would naturally expect.
> >
> > Furthermore, when comparing the proposed method to prior work like Hallo3, the changes, removing convolutions and feeding all audio tokens to each frame, appear to be straightforward design choices, as this approach is already widely used in other UNets. From my perspective, the decision of whether to downsample the audio tokens seems closer to an implementation detail for an ablation study, rather than a fundamental innovation.

---

> ### Author Response · Authors · 2025-08-04
> **Discussion (1/2)**
>
> Dear Reviewer `qDos`,
>
> We are **encouraged** that you find our response has resolved most of your concerns, and we **greatly appreciate** the opportunity to continue this discussion.
>
> ## On Multi-stage Training:
>
> Thank you for your agreement with our proposed strategy — “Given the impressive results, I agree that this is a fair approach.”  To further clarify its necessity. We analyzed Hallo3’s training data captions. Among the 101,543 captions used in Hallo3, 33,909 (33.39%) exactly match the phrase "a person is talking." In contrast, Mocha’s training strategy provides detailed captions including emotions, body language, actions, and camera movements and multiclip descriptions. We also consistently sample 20% of the data from a diverse text-to-video dataset. This helps Mocha unlock the ability to control the person beyond a frozen pose or a fixed camera-facing view — enabling movement and natural interaction with the environment and objects that are not present in audio-annotated data.
>
> ## On the Localized Audio Attention:
>
> We would like to clarify the motivation and novelty of our design.
>
> **The core challenge we address** is the mismatch in resolution between audio and video tokens in DiT-based models.
> Suppose the video frames $f_1,f_2,f_3,f_4$ correspond to audio features $\alpha_1,\alpha_2,\alpha_3,\alpha_4$ respectively:
>
> $f_1↔\alpha_1$;
> $f_2↔\alpha_2$;
> $f_3↔\alpha_3$;
> $f_4↔\alpha_4$
>
> For EMO, which is built on a 2D U-Net and does not compress frames, there is no mismatch issue — each frame aligns directly with its corresponding audio feature.
> Previous U-Net-based approaches studied whether to augment $\alpha_i$ with adjacent features —just as the reviewer mentioned, “each frame attends to audio tokens near its timestamp”:
>
> $f_1$ ↔ {$\alpha_1,\alpha_2$}
>
> $f_2$ ↔ {$\alpha_1,\alpha_2,\alpha_3$}
>
> However, in DiT-based models, due to 3d vae, 4 or 8 adjacent frames are compressed into a single video token:
> $f_1,f_2,f_3,f_4→x_1$
>
> This breaks fine-grained alignment. Our study focuses on how to recover the fine-grained alignment, e.g., how $f_1 ↔ \alpha_1$ can still be preserved under this compression.
>
> Essentially, the question becomes:
> How do we align {$x_1,x_2, …$} to {$α_1,α_2,α_3,α_4,α_5,α_6,α_7,α_8, …$}?
>
> This challenge is different from the one addressed in EMO, which explores whether $f_i$ should be augmented with features from $f_{i+1}$.
>
> This is a  **critical question** for dialogue-driven video generation using DiT, as most modern video generation models are based on DiT. However, it has not been thoroughly explored in existing work
>
> We raise the following research questions:
> 1. Should we compress the audio features to enforce one-to-one alignment (as in Hallo3)?
>
>     {$α_1,α_2,α_3,α_4$} → $α_1*$ via convolution.
>
>     then $x_1$ and $α_1*$ are directly aligned i.e. $x_1↔α_1*$
> 2. Should we keep audio features in their original resolution? (Naive Audio Attention in Appendix)
>
>     {$x_1,x_2, …$} ↔ {$α_1,α_2,α_3,α_4,α_5,α_6,α_7,α_8,…$}
> 3. If we keep the audio features in original resolution:
>    - Does positional embedding help alignment? (Audio Attention with Positional Embedding in Appendix)
>
>      Ideally, as in LLMs, tokens can learn where to attend.
>      Can the $x$ tokens similarly learn to use positional information to extract relevant audio tokens?
>      Which type of positional embedding is more effective — absolute (sinusoidal), learnable, or relative (RoPE)?
>
>      {$x_1$} ↔ learns to attend to {$α_1,α_2,α_3,α_4$}
>
>      {$x_2$} ↔ learns to attend to {$α_5,α_6,α_7,α_8$}
>    - Does a strong inductive bias like windowed attention help alignment?
>      (Our proposed method)
>
>      {$x_1$} ↔ attends only to {$α_1, α_2, α_3, α_4$}
>
>      {$x_2$} ↔ attends only to {$α_5, α_6, α_7, α_8$}
>
> Our findings show that the proposed window-based cross-attention achieves the best results among all variants.
> This focus is valuable for future research on building dialogue-driven video generation models in the DiT framework.
> Our study carefully and systematically explores many previously unaddressed directions and provides important insights and lessons for future work.

---

> ### Author Response · Authors · 2025-08-05
> **Discussion (2/2)**
>
> ## On the Novelty of MoCha:
>
> We would also like to highlight the novelty of the **MoCha framework itself** and also draw attention to **Section 3.3: Multi-character Turn-based Conversation**.
>
> 1) MoCha framework is novel as it targets the new task: text+audio2video generation. Distinct from all existing work relying on separate stages or auxiliary inputs, MoCha's promising results demonstrate that vivid talking characters can be generated from scratch directly using text to define a scene and audio to control facial movements. We don't need a pre-generated first frame or hand pose. We don't need any facial masks during training. This flexibility provides new insight to the community.
>
> 2) In **Section 3.3: Multi-character Turn-based Conversation**. We introduce a novel method that employs **character-tagged prompts** and **self-attention across video tokens** to **generate multi-clip in parallel** and enables, for the first time **multi-speaker interaction**.
> This is a critical component for generating complex, movie-style scenes, as turn-based conversations between multiple characters are essential in cinematic storytelling— a novel capability not achievable with prior stage-based or mono-speaker approaches.
>
> In addition to
>
> 3) MoCha's strong audio-video synchronization results from our careful and unique study on aligning video and audio tokens within the DiT framework. Our novel window-based audio attention mechanism makes this synchronization possible.
>
> 4) MoCha achieves universal controllability via natural languages and realistic character movements through a novel designed training strategy.

---

> > ### Comment · Reviewer_qDos · 2025-08-05
> >
> > Thank you for the detailed explanation.
> >
> > Although the authors have investigated many variants, the finally chosen method is still similar to the well-established cross-attention in U-Net. It reduces the use of audio features from neighboring timesteps, but considering that wav2vec itself already aggregates neighborhood features, I do not believe the 'window-based cross-attention' is novel. Window-based approaches have long been commonly used, and 'Localized Audio Attention' is an existing paradigm.
> >
> > Therefore, I suggest that the authors reconsider the wording in this section and instead focus the main contribution on the multi-clip and multi-turn generation, which has shown impressive results.

---

> > > ### Author Response · Authors · 2025-08-06
> > >
> > > Dear Reviewer `qDos`,
> > >
> > > We sincerely thank you for the thoughtful and constructive feedback. Your comments were instrumental in helping us improve the clarity and quality of the paper. We are encouraged that you found our responses have resolved most of your concerns.
> > >
> > > We respectfully invite you to re-evaluate the manuscript in light of the discussion and clarifications we have provided.
> > >
> > > Please feel free to reach out with any further suggestions or concerns.
> > >
> > > Best regards,
> > > The Authors

---

> > > > ### Comment · Reviewer_qDos · 2025-08-07
> > > >
> > > > Thank you for the clarification. I have re-examined the paper and considered the comments from other reviewers. It appears there is a consensus regarding the novelty of the window-based audio conditioning. I believe the core contribution of this work, demonstrating that effective multi-character, turn-based conversational video generation can be achieved through joint data training and the removal of certain conditions, is meaningful. However, the claims regarding the novelty of the implementation methods, such as the window-based audio conditioning require revision. I am fine with the other parts of the paper.
> > > >
> > > > Considering the authors will revise the description, I have raised my score.

---

> ### Author Response · Authors · 2025-08-05
>
> Dear Reviewer `qDos`,
>
> Thank you for your suggestion and thoughtful discussion! In the final version, we will revise the abstract, introduction and method sections to focus the main contribution on the multi-clip and multi-turn generation according to the suggestions.

---

### Official Review · Reviewer_gs1v · 2025-06-29

**Clarity:** 3
**Significance:** 3
**Originality:** 2
**Rating:** 5
**Confidence:** 3

**Summary:**

Mocha proposed a DiT-based text- and speech-driven Talking Character generation method, which uses cross attention and localized audio attention to inject text and speech conditions, achieving state-of-the-art performance in Talking Character generation.

**Questions:**

See weakness.

**Ethical Concerns:**

["NO or VERY MINOR ethics concerns only"]

**Final Justification:**

I recommend accepting this article.

**Limitations:**

As mentioned in weakness, this method is based on a closed-source model and the experimental overhead is too high to be reproduced.

**Paper Formatting Concerns:**

None.

**Quality:**

3

**Strengths And Weaknesses:**

**Strengths:**

1. Compared to other methods, this approach achieves video generation that is better aligned with both text and speech, marking a significant step toward the practical application of video generation.
2. The use of localized audio attention enables better alignment between lip movements and speech in the generated videos, making it a sensible and effective design.

**Weaknesses:**

1. The method has not been open-sourced, nor are there plans for open-sourcing it. Additionally, the experiments were conducted using Meta’s internal MovieGen framework, making it difficult for the community to reproduce the results.
2. The five steps mentioned in Curriculum Training lack ablation studies for each stage, making it seem unnecessary to design such a complex process.
3. The method has high resource consumption. Despite being based on a powerful video generation model, it still requires 64 computing nodes for post-training. It does not consider freezing most of the network modules during training or using other efficient fine-tuning methods.

---

> ### Author Rebuttal · Authors · 2025-07-31
>
> Dear reviewer `gs1v`,
>
> We sincerely appreciate the time and effort you have dedicated to reviewing our work and providing valuable feedback. Below, we are pleased to share our detailed responses to your comments.
>
> **W1 The method has not been open-sourced, nor are there plans for open-sourcing it.**
>
> As authors, we are fully committed to open-source research.
> We have implemented MoCha on top of the open-source backbone Wan2.1 and will include a link to the code in the final revision.
> We have already included technical details such as model parameters, the full data processing pipeline, and example captions in the submission. We will further enrich the final version with as many details as possible to benefit the community.
>
> **W2 The five steps mentioned in Curriculum Training lack ablation studies for each stage, making it seem unnecessary to design such a complex process.**
>
> **Answer:** Thank you for the suggestions. Below, we provide both explanation and supporting results to justify the necessity of our curriculum design.
>
> Traditional talking-head generation tasks typically use cropped face videos with limited variability. In contrast, MoCha tackles a significantly more complex task, involving: talking faces→half-body gesture→full-body movements→camera control→multi-character, multi-clip. As the visual scope increases, the mouth region becomes smaller and harder to supervise. Training on all data types simultaneously is highly inefficient for learning audio-driven control—especially since MoCha does not rely on strong control signals like reference frames. To tackle this, our Shot-Type-Based Curriculum is used to structures the learning process based on shot complexity. This method significantly improves convergence.
>
> We provide an ablation study of different stages using the 4B version, trained for the same number of steps (20K) and evaluated on MoChaBench.  Stage 0 corresponds to T2V pretraining, and Stage 1 introduces the audio signals, which are indispensable for the task. Therefore, our ablation focuses on the contributions of Stage 2 and Stage 3
>
> | Method                                                       |Sync-C↑| Sync-D ↓| Facial Expression | Action Naturalness | Text Alignment | Visual Quality |
> |-                                                             | -     | -       |-                  |-                   |-               | -              |
> | DiT-4b + (close-up and medium close up curriculum training)  | 5.718 | 8.374   | 2.96              | 1.37               | 1.18           | 2.67           |
> | DiT-4b + (close-up and medium close up joint training)       | 3.846 | 10.378  | 1.25              | 1.16               | 1.06           | 2.55           |
>
> Analysis: Medium close-up data introduces information such as co-speech gesture and the character's overall motion. Joint training on close-up and medium close-up data shows significantly degraded results in lip-sync metrics compared with curriculm training, as the facial region in medium close-up data is relatively smaller, providing limited supervision for the audio-conditioning module and leading to slower learning.
>
> | Method                                                       |Sync-C↑| Sync-D ↓| Facial Expression | Action Naturalness | Text Alignment | Visual Quality |
> |-                                                             | -     | -       |-                  |-                   |-               | -              |
> | DiT-4b + (close-up, medium close-up and medium shot curriculum training)  | 5.692 | 8.403   | 2.92      | 2.87       | 2.94           | 2.69           |
> | DiT-4b + (close-up, medium close-up and medium shot joint training)       | 2.394 | 11.291  | 1.22      | 2.66       | 1.58           | 2.39           |
>
> Analysis: Medium shot data focuses on full-body actions and camera control.
> Joint training with all shot types at once further degrades lip-sync performance and text alignment. In contrast, curriculum training gradually builds the model’s ability to handle increasing complexity, resulting in balanced performance across metrics. While lip-sync scores are slightly lower than the previous stage, gains in action naturalness and text alignment are substantial.
> These ablation studies collectively demonstrate the necessity and effectiveness of each stage in our proposed curriculum.
>
>
> **W3: Consider freezing most of the network modules during training. L1: The experimental overhead is too high**
>
> **Answer:** We thank the reviewer for raising this important question.
>
> To address concerns about high resource consumption,
> we provide the following experiment to show that our proposed training strategy and localized audio attention methods work for small models.
> The base models of Hallo3 and MoCha-4b have similar sizes and comparable initial performance.
> After learning the audio condition, MoCha-4b outperforms Hallo3 across all axes. Hallo3 suffers from worse generalization, as evidenced by its degraded performance in text alignment and action naturalness compared to its base model.
> | Method                         |Sync-C↑| Sync-D ↓| Facial Expression | Action Naturalness | Text Alignment | Visual Quality |
> |-                               | -     | -       |-                  |-                   |-               | -      |
> | CogVideoX-5b (Base for Hallo3) | N/A   | N/A     | 2.88              |  2.85              | 2.91           | **2.83**       |
> | Hallo3                         | 4.866 | 8.963   | 2.25 (-0.63)      |  2.13 (-0.72)      | 2.35 (-0.56)   | 2.36 (-0.47)   |
> | T2V Base for MoCha-4b          | N/A   | N/A     | 2.86              |  2.76              | 2.93           | 2.71           |
> | MoCha-4b                | **5.692** | **8.403**  | **2.92** (+0.06)  | **2.87** (+0.11)   | **2.94**(+0.01)| 2.69 (−0.02)   |
>
> This demonstrates that the effectiveness of our approach is not limited to large models requiring high resource consumption.
>
> Our research goal—as stated in the claims—is to **bridge the gap between generic video generation and constrained talking-head synthesis**, which requires a promising talking character model that can sync well with speech and perform actions naturally with overall good visual quality.
>
> However, as shown in the table above, base models with 4B/5B parameters still fail to consistently generate videos with a score of 3, meaning they cannot reliably produce mostly artifact-free and prompt-following results.
> Therefore, to support our goal, we conduct the research on top of a large base model.
>
> We have also conducted experiments during research where we only trained the newly introduced audio attention module while freezing the remaining backbone.
> We provide the results as follows:
> | Method                          |Sync-C↑| Sync-D ↓ | Facial Expression | Action Naturalness | Text Alignment | Visual Quality |
> |-                               | -     | -       |-                  |-                   |-               | -      |
> |MoCha                                        | 6.037 | 8.103    | 3.82              | 3.82               | 3.85           | 3.72 |
> |DiT + Train Audio Cross Attention Only       | 1.859 | 12.784   | 1.04              | 1.00               | 1.17           | 1.21           |
> |DiT                                          | N/A   | N/A      | 3.49              | 3.82      | 3.16          | 3.73       |
>
> We observed a dramatic degradation in model performance across all axes when training only the audio cross-attention module, compared with full training (MoCha) for the same number of steps.
> This observation differs from talking-head tasks, where training only the newly added module is often more effective than full fine-tuning.
> We hypothesize that since talking-head tasks always include a first-frame/pose condition—a strong conditioning signal—they converge more easily.
> In contrast, talking character generation is a more complex task.
> Thus, we consider full training of the model to be more effective in our setting.
>
> The resulting model, MoCha, is **among the first to demonstrate that movie-grade storytelling can be achieved without relying on auxiliary prior signals**, and **the first to support multi-character, turn-based dialogue clip generation**, which **bridges the gap between generic video generation and constrained talking-head synthesis**.
> Given the promising results of MoCha, we believe it paves the way for future research on smaller yet stronger models (possibly distilled from large models) to extend talking character generation to broader applications.

---

> > ### Comment · Reviewer_gs1v · 2025-08-04
> > **Raise Score**
> >
> > Thanks to the author for the detailed answer, which basically solved my doubts. I look forward to seeing your work in the open source community one day. I decided to raise my score to 5 points.

---

### Official Review · Reviewer_oyFP · 2025-06-30

**Clarity:** 4
**Significance:** 4
**Originality:** 4
**Rating:** 6
**Confidence:** 5

**Summary:**

The paper presents a new complex task of talking character generation, i.e. generating lip-synced video of full portrait of one or more characters beyond just the facial region. Given it is a new task, the paper also presents a new benchmark and method for the same. The method is novel for four reasons: 1) The method does not use any additional signals apart from the coarsely aligned speech and video description text 2) Uses windowed attention between video and audio instead of full cross-attention 3) Uses curriculum-based training across two axes - the modality and task complexity 4) able to generate coherent multi-character clips in parallel. Subjective (human rating on scale 1-4) and Objective results (lip-sync metrics) clearly state the effectiveness of the method. The benchmark for talking character generation contains 200 instances of human verified text, speech and video.

**Questions:**

1. The authors have presented three good limitations in the appendix. What are some possible ways to resolve these problems and further improve the method?
2. What is the total number of training hours and what kind of GPUs were used for training?

**Ethical Concerns:**

["NO or VERY MINOR ethics concerns only"]

**Final Justification:**

The authors have resolved my questions. I would like to maintain my high score. The work has strong evaluation and high impact on the area of video generation. Some reviewers have questioned the novelty and results of the work, but I found the claims in paper to be well supported and the results are solid.

**Limitations:**

yes

**Paper Formatting Concerns:**

The table captions should be on the top of the table.

**Quality:**

4

**Strengths And Weaknesses:**

Strengths:
1. Introduces new task, new benchmark and new method for talking character generation.
2. Limitation section is well thought out.
3. The appendix is detailed and contains good ablation studies and subjective evaluation details
4. Writing is clear and good.
5. The proposed method clearly outperforms existing methods both in subjective and objective evaluations.

Weaknesses:
1. Please include some sections from appendix to main work in the final version
2. No mention of intention to open source the code, data and checkpoints. Though this is not compulsory but recommended.

---

> ### Author Rebuttal · Authors · 2025-07-31
>
> Dear reviewer `oyFP`,
>
> We sincerely appreciate the time and effort you have dedicated to reviewing our work and providing valuable feedback.
>
> We are encouraged that you find our paper to be "well written" and our proposed methods to be **significant and novel**."
>
> Below, we are pleased to share our detailed responses to your comments.
>
> **W1 Please include some sections from the appendix to the main work in the final version**
>
> Thank you for the suggestion. In the final version, we will move the Section A ablations from the appendix to the main paper to ensure clarity and completeness
>
> **W2 No mention of intention to open source the code, data and checkpoints. Though this is not compulsory but recommended.**
>
> As authors, we are fully committed to open-source research.
> We have implemented MoCha on top of the open-source backbone Wan2.1 and will include a link to the code in the final revision.
> We have already included technical details such as model parameters, the full data processing pipeline, and example captions in the submission. We will further enrich the final version with as many details as possible to benefit the community.
>
> **Q1 The authors have presented three good limitations in the appendix. What are some possible ways to resolve these problems and further improve the method?**
>
> For the limitation: "If the caption is too vague and doesn't describe the facial attribute or shot type, the model may generate a wide shot where the character is far from the camera, making lip sync difficult to observe."
> We can fine-tune the model via SFT on a curated dataset. We could collect a dataset of (caption, preferred shot framing) pairs where vague captions are matched with preferred video compositions (e.g., close-up shots for talking scenes). Use this to fine-tune the model to better infer appropriate shot types even when explicit cues are missing. For instance, given "A man playing skateboard at a skatepark" with accompanying speech, the model could learn to prioritize closer shots for improved lip sync alignment.
> We could also deploy annotators to rate generated videos on lip sync clarity and framing quality. Use these ratings to train a reward model and apply reinforcement learning or preference-based learning to optimize generation toward human-preferred outputs. This would teach the model to implicitly prioritize facial visibility when speech is present, even from vague captions.
>
> For the limitation : "When multiple characters appear in the same scene, sometimes the lip-sync quality is degraded when the character is far from the camera, potentially due to limited training data."
> When multiple characters are present, the model may struggle to identify which character the speech corresponds to, based solely on the text. To mitigate this, we could incorporate character embeddings alongside speech embeddings to provide clearer grounding.  For example, a set of learnable embeddings could be added to the speech embeddings to indicate the target character. This may help the model better associate the audio with the correct character and improve lip sync quality, particularly in multi-character scenes.
>
> For the limitation :"When increasing the speech CFG from the default value of 7.5 to 12, the model tends to generate overly expressive characters. For example: 'A tracking shot circling around the man as he ties a tie over his blue suit. He speaks to the camera while adjusting the knot, maintaining eye contact throughout.'" To mitigate this, we could rescale the norm of the guided prediction at each denoising step to match the original conditional signal. This helps prevent exaggerated behaviors caused by excessively high CFG values.
>
> **Q2 What is the total number of training hours and what kind of GPUs were used for training?**
>
> Our experiments were conducted on H100 GPUs, totaling 49,152 GPU hours.

---

> > ### Comment · Reviewer_oyFP · 2025-08-04
> >
> > Thank you for the detailed responses. I would like to maintain my current rating.

---

### Official Review · Reviewer_Mvje · 2025-07-02

**Clarity:** 3
**Significance:** 3
**Originality:** 2
**Rating:** 4
**Confidence:** 4

**Summary:**

This paper presents an end-to-end video generation model that synthesizes talking characters from only text and audio inputs. It goes beyond facial regions to include full-body motions, multi-character interactions and environmental dynamics. To this end, the authors modify the text-to-video DiT model and inject audio conditions through temporally aligned audio cross-attention. A curriculum-based multimodal training strategy is introduced to combine speech-labeled and text-only data to improve generalization. Experiments show that the proposed method support coherent, turn-based dialogue among multiple characters, outperforming baselines in lip-sync quality, expression naturalness, and text alignment.

**Questions:**

None

**Ethical Concerns:**

["NO or VERY MINOR ethics concerns only"]

**Final Justification:**

Thanks for the detailed response, which addresses most of my concerns. After reading the rebuttal and other reviews (as well as the discussion), I would like to maintain my initial positive scores. I am glad to see that the authors have implemented MoCha on top of the open-source backbone Wan2.1 and promise to release it in the revision.

**Limitations:**

yes

**Paper Formatting Concerns:**

no formatting concerns

**Quality:**

3

**Strengths And Weaknesses:**

### Strengths
* Unlike traditional talking-head generation (limited to facial regions and single speakers) , this paper explicitly targets full-body generation with multi-character interactions. This framing fills a clear gap in automating cinematic storytelling and content creation,

* Human evaluations across five axes (lip-sync, expression naturalness, action alignment, etc.) show that the proposed method outperforms existing baselines (SadTalker, Hallo3) by a large margin.

* The paper is overall well-written, and the technical details is clearly provided in the appendix.

### Weaknesses
* The so-called "Localized Audio Attention" seems straight-forward and is not novel to me. Injecting audio condition through temporally-aligned cross-attention has already been explored in Hallo3 [27]. It would be better if the authors could provide more discussion on the difference between this method and Hallo3 in terms of the design of audio injection modules.

* The curriculum-based multimodal training strategy is also not novel enough; a similar idea has already been proposed in OmniHuman [20], which also introduces a multi-stage training strategy and follows a curriculum progressing from text to audio. Unfortunately,  this paper does not explicitly address the connections and distinctions between its strategy and that of OmniHuman.

* The baselines in the comparison experiments rely on reference image inputs. However, it is unclear how the reference images are selected for these baselines in the comparison. In addition, it is unclear how the proposed method could be adapted to reference-conditioned settings.

---

> ### Author Rebuttal · Authors · 2025-07-31
>
> Dear reviewer `Mvje`,
>
> We sincerely appreciate the time and effort you have dedicated to reviewing our work and providing valuable feedback. Below, we are pleased to share our detailed responses to your comments.
>
> **W1 Localized Audio Attention seems has been explored in Hallo3**
>
> To address the reviewer’s concern, we present a design comparison between Hallo3 and MoCha.
> Hallo3 applies stacked convolutions twice to temporally compress audio features, aligning them with the 3D VAE’s compression ratio used for video frames. However, we find that such compression degrades fine-grained audio details. In real-life, dialogue-driven movie generation scenarios, mouth movements can be extremely rapid, and accurately reproducing phonemes such as /iː/, /ai/, /tʃ/, and /θ/ in quick succession requires high-resolution audio features.
> To preserve these fine-grained details, MoCha avoids temporal compression of audio features. Instead, we propose to constrain each video token to attend to a local window of high-resolution audio tokens, enabling precise audio-visual alignment while retaining rich temporal fidelity in the audio signal.
>
> We provide a direct comparison of both methods on a DiT-T2V-4B backbone:
>
> |Method |Sync-C↑| Sync-D ↓|
> |-|-|-|
> |DiT-4B + Localized Audio Attention (MoCha-4B)|  5.692 |8.403|
> |DiT-4B + Hallo3 Audio Attention|  4.885 |8.896|
>
> Key Observation: Our novel localized audio attention approach results in superior audio-visual synchronization. Additionally, as shown in visual demos, Hallo3 often exhibits articulation errors and lip-sync artifacts when generating challenging phonemes like /iː/, /ai/, /tʃ/, and /θ/.  In contrast, MoCha captures fine-grained articulation accurately and surpasses all baselines by a large margin.
>
> Furthermore, we acknowledge that Hallo3(Section 3.2) studies 3 audio-injection strategies: (1) self-attention, (2) adaptive-norm, and (3) cross-attention, with cross-attention proving to be the most effective. This is indeed a valuable insight! However no existing work thoroughly studies how to effectively align audio features with temporally compressed video tokens within the cross-attention framework.  Our work addresses this gap directly. As detailed in Appendix Section A.1, we conduct a comprehensive ablation study comparing:(1) Naive Audio Cross Attention (2) Naive Cross-Attn with various positional embeddings (RoPE, learnable, sinusoidal) (3) Our proposed localized audio attention. Our method consistently outperforms all variants. This study provides valuable useful for future research on audio-visual alignment.
>
> **W2 Address the distinctions between proposed curriculum-based multimodal training strategy and that of OmniHuman**
>
> We provide an explanation below, followed by supporting results.
>
> Our training strategy contains two parts **Shot-Type-Based Curriculum** + **Mixed-Modal Sampling**.
>
> We agree that existing works(omnihuman) do use multi-stage training. However, they use it to add auxiliary signals, gradually introducing them (face image → audio → pose) to the model in different stages (omnihuman Section 4.1). At the audio stage, they simply feed all audio-annotated data into the model. This works fine as these models' tasks are talking heads generation, and the data are mostly cropped human face videos with similar face sizes. But for MoCha, our task and data are much more complex, involving talking faces→half-body gesture→full-body movements→camera control→multi-character, multi-clip. As the visual scope increases, the mouth region becomes smaller and harder to supervise. Training on all data types simultaneously to learn audio control is clearly inefficient, especially since MoCha doesn't use strong control signals like reference frames. To tackle this, our **Shot-Type-Based Curriculum** is not used to add new signals like existing work but instead structures the learning process based on shot complexity.  **This method significantly improves convergence, as shown in Appendix Table 5.**
>
> SoTA methods(including hallo3, omnihuman) struggle with generating talking characters that go beyond static poses and fixed camera views. They fail to generalize to realistic cinematic scenarios, such as characters talking while expressing emotions, naturally performing actions, interacting with objects, or moving alongside the camera in response to text conditions. We identify two reasons (1) They overlook the richness of text conditions, often reducing prompts to oversimplified inputs like “a person is talking.” (2)They rely exclusively on human data, sampled primarily from audio/pose-driven datasets. To directly address this, we propose **Mixed-Modal Sampling**: We always provided high-quality and multi-grained captions to the model (as shown in Figure 2); During training we consistently sample 20/% of the data from a diverse text2video dataset.
> **This method enables MoCha to achieve superior generalization** in scenarios involving complex lighting, human-animal interactions, and unlock the universal controllability via prompts, eventually **unlock the multiple-character turned based converstation**-**a capability not supported by any existing method**.
>
> We provide the following experiment against the strongest baseline Hallo3.
> The base models of Hallo3 and MoCha-4b have similar size and on-pair performance initially.
> After learning the audio condition, MoCha-4b outperforms Hallo3 on all axes. Hallo3 suffers from worse generalization, as evidenced by its degraded performance in text alignment and action naturalness compared to its base model.
> This highlights the effectiveness of our proposed training strategy and underscores the uniqueness of our contribution.
> | Method |Sync-C↑| Sync-D ↓| Facial Expression | Action Naturalness | Text Alignment | Visual Quality |
> |- | - | - |- |- |- | - |
> | CogVideoX-5b (Base for Hallo3) | N/A   | N/A     | 2.88  |  2.85 | 2.91 | **2.83**  |
> | Hallo3 | 4.866 | 8.963   | 2.25 (-0.63) |  2.13 (-0.72)  | 2.35 (-0.56)   | 2.36 (-0.47) |
> | T2V Base for MoCha-4b | N/A | N/A | 2.86   |  2.76  | 2.93 | 2.71  |
> | MoCha-4b | **5.692** | **8.403**  | **2.92** (+0.06)  | **2.87** (+0.11)   | **2.94**(+0.01)| 2.69 (−0.02) |
>
> **W3 It is unclear how the reference images are selected for these baselines in the comparison**
>
> **Answer:**  We would like to emphasize that our work is dedicated to the novel task: **talking characters**.
> The experiments were designed to compare with existing methods that can only do talking head generation, which are all I2V models. To follow their inference settings, we used the first frame of the video generated from MoCha as input. We manually inspect each frame and avoid blurry or artifacts. The frames also contain clear facial details, making them suitable as reference images. We then crop and resize the frame to match each baseline's training distribution and ensure the head region is centred.
> Despite these efforts, we acknowledge that the first frame from MoCha may still fall outside the baselines’ training distributions. To address this, we conducted further experiments using three groups of inputs:
>
> Group 1: 3–5 real images from the Internet matching the prompts (e.g., “woman driving a car”), with clear head poses and scene alignment.
>
> Group 2: 5 synthetic images generated by CogView4, GPT-Image-1, and FLUX.1-dev.
>
> Group 3: Original MoCha-generated frames.
>
> These groups test the robustness of baselines across diverse inputs.
> | Method  |Sync-C↑| Sync-D ↓| Facial Expression | Action Naturalness | Text Alignment | Visual Quality |
> |-|-|-|-|-|-|-|
> | Hallo3 (Group1)  | 4.820 | 8.899 | 2.33   | 2.33 | 2.35  | 2.32  |
> | Hallo3 (Group2)  | 5.024 | 9.263 | 2.38  | 2.35 | 2.38 | 2.36  |
> | Hallo3 (Group3)  | 4.866 | 8.963 | 2.25  | 2.13 | 2.35  | 2.36 |
> | SadTalker (Group1) | 4.824 | 9.133   | 1.35 | 1.00 | N/A  | 2.98|
> | SadTalker (Group2) | 4.569 | 9.310   | 1.04  | 1.00 | N/A  | 2.93  |
> | SadTalker (Group3) | 4.727 | 9.239   | 1.14 | 1.00 | N/A   | 2.95  |
> | AniPortrait (Group1)| 1.749 | 11.005  | 1.17 | 1.00 | N/A   | 1.46 |
> | AniPortrait (Group2)| 1.688 | 12.084  | 1.13 | 1.00 | N/A  | 1.45 |
> | AniPortrait (Group3)| 1.740 | 11.383  | 1.12 | 1.00 | N/A | 1.45 |
> | MoCha |**6.037**|**8.103** | **3.82**  | 3.82 | **3.85** | 3.72 |
>
> **Key Observations: (1)Baseline performances are robust across groups, showing MoCha-generated frames are not out-of-domain for the baselines. (2)MoCha significantly outperforms all baselines, confirming its superiority is not due to biased inputs**.
>
> **W4 How the proposed method could be adapted to reference-conditioned settings.**
>
> Reference-conditioned generation is a relatively well-studied area, with many existing methods achieving strong results. MoCha can be extended using approaches like FullDiT—encoding the reference image via the VAE and concatenating it with video tokens for 3D self-attention processing, or following the Movie Gen's Personalization method, concatenating the reference image embedding to the text embedding and further being processed through cross-attention.
> Meanwhile, dialogue-driven video generation area was limited to talking-head synthesis, the main research focus of this paper is to **bridge the gap between generic video generation and talking-head synthesis, making cinematic storytelling possible**. We introduced a new task called talking characters, along with a corresponding benchmark and a promising model. MoCha is among the first to demonstrate that movie-grade storytelling can be achieved without relying on auxiliary prior signals, and the first to support multi-character, turn-based dialogue clip generation. Given the promising results of MoCha, we believe it paves the way for future research on unified models that generate talking characters guided by reference images.

---

> > ### Comment · Reviewer_Mvje · 2025-08-07
> >
> > Thanks for the detailed response, which addresses most of my concerns. After reading the rebuttal and other reviews (as well as the discussion), I would like to maintain my initial positive scores. I am glad to see that the authors have implemented MoCha on top of the open-source backbone Wan2.1 and promise to release it in the revision.  Please do not just publish a coming soon link.

---

> > > ### Author Response · Authors · 2025-08-07
> > >
> > > Dear Reviewer `Mvje`,
> > >
> > > Thank you for your thoughtful feedback, and we are encouraged that you have maintained a positive assessment. If there are any remaining concerns or points that you feel could be clarified or strengthened, we would be happy to address them further.
> > >
> > > Best regards,
> > >
> > > The Authors

---

> ### Author Response · Authors · 2025-08-06
>
> Dear Reviewer `Mvje`,
>
> We sincerely appreciate the time and effort you have dedicated to reviewing our work, as well as your constructive comments.
>
> We are encouraged that you found our paper **fills a clear gap in automating cinematic storytelling and content creation.**
>
> We have submitted point-by-point responses to your comments. As the discussion phase is nearing its end. We hope our responses have addressed your concerns, and we would be happy to clarify anything further if needed.
>
> Best regards,
>
> The Authors

---

### Comment · Area_Chair_wqz1 · 2025-08-01

Dear reviewers,

Thank you for your time and effort in reviewing this submission. We are in the phase of author-reviewer discussion until 6th August.

I highly encourage you to discuss with the authors for clarification of your concerns. Your active participation in the discussion will be the main guarantee for high-quality publications in our community.

We now have a bit of mixed reviews for this paper submission. I hope you could read the rebuttal and comments from each other for a respectful discussion, please. Thank you!

Best regards,\
Your AC

---

### Decision · Program_Chairs · 2025-09-17

**Decision:**

Accept (spotlight)

**Comment:**

This paper received overwhelmingly positive feedback from all reviewers. It presents a text-to-video generation model that supports full-body motions, multi-character interactions, and environmental dynamics. This represents a novel and meaningful task. The alignment of speech and video is particularly appreciated. The method demonstrates significantly better performance than baseline approaches in both subjective and objective evaluations. With the authors’ commitment to releasing the source code, this work has the potential to make a high impact on the video generation community.

The reviewers raised some concerns regarding the experimental setup and the degree of novelty. The authors provided thorough clarifications during the rebuttal period. Taking all of this into account, I recommend acceptance.